

# Resurrecting a subgenus to genus: molecular phylogeny of *Euphyllia* and *Fimbriaphyllia* (order Scleractinia; family Euphylliidae; clade V)

Katrina S. Luzon[1,2,3,*], Mei-Fang Lin[4,5,6,*], Ma. Carmen A. Ablan Lagman[1,7], Wilfredo Roehl Y. Licuanan[1,2,3] and Chaolun Allen Chen[4,8,9,*]

[1] Biology Department, De La Salle University, Manila, Philippines
[2] Shields Ocean Research (SHORE) Center, De La Salle University, Manila, Philippines
[3] The Marine Science Institute, University of the Philippines, Quezon City, Philippines
[4] Biodiversity Research Center, Academia Sinica, Taipei, Taiwan
[5] Department of Molecular and Cell Biology, James Cook University, Townsville, Australia
[6] Evolutionary Neurobiology Unit, Okinawa Institute of Science and Technology Graduate University, Okinawa, Japan
[7] Center for Natural Sciences and Environmental Research (CENSER), De La Salle University, Manila, Philippines
[8] Taiwan International Graduate Program-Biodiversity, Academia Sinica, Taipei, Taiwan
[9] Institute of Oceanography, National Taiwan University, Taipei, Taiwan
[*] These authors contributed equally to this work.

Corresponding authors
Katrina S. Luzon,
katrina.luzon@dlsu.edu.ph
Chaolun Allen Chen,
cac@gate.sinica.edu.tw

## ABSTRACT

**Background**. The corallum is crucial in building coral reefs and in diagnosing systematic relationships in the order Scleractinia. However, molecular phylogenetic analyses revealed a paraphyly in a majority of traditional families and genera among Scleractinia showing that other biological attributes of the coral, such as polyp morphology and reproductive traits, are underutilized. Among scleractinian genera, the *Euphyllia*, with nine nominal species in the Indo-Pacific region, is one of the groups that await phylogenetic resolution. Multiple genetic markers were used to construct the phylogeny of six *Euphyllia* species, namely *E. ancora*, *E. divisa*, *E. glabrescens*, *E. paraancora*, *E. paradivisa*, and *E. yaeyamaensis*. The phylogeny guided the inferences on the contributions of the colony structure, polyp morphology, and life history traits to the systematics of the largest genus in Euphylliidae (clade V) and, by extension, to the rest of clade V.

**Results**. Analyses of cytochrome oxidase 1 (*cox1*), cytochrome b (*cytb*), and β-tubulin genes of 36 colonies representing *Euphyllia* and a confamilial species, *Galaxea fascicularis*, reveal two distinct groups in the *Euphyllia* that originated from different ancestors. *Euphyllia glabrescens* formed a separate group. *Euphyllia ancora, E. divisa, E. paraancora, E. paradivisa*, and *E. yaeyamaensis* clustered together and diverged from the same ancestor as *G. fascicularis*. The 3′-end of the *cox1* gene of *Euphyllia* was able to distinguish morphospecies.

**Discussion**. Species of *Euphyllia* were traditionally classified into two subgenera, *Euphyllia* and *Fimbriaphyllia*, which represented a dichotomy on colony structure. The paraphyletic groups retained the original members of the subgenera providing a strong basis for recognizing *Fimbriaphyllia* as a genus. However, colony structure was found

to be a convergent trait between *Euphyllia* and *Fimbriaphyllia*, while polyp shape and length, sexuality, and reproductive mode defined the dichotomy better. Species in a genus are distinguished by combining polyp morphology and colony form. The cluster of *E. glabrescens* of the *Euphyllia* group is a hermaphroditic brooder with long, tubular tentacles with knob-like tips, and a phaceloid colony structure. The *Fimbriaphyllia* group, with *F. paraancora, F. paradivisa, F. ancora, F. divisa*, and *F. yaeyamaensis*, are gonochoric broadcast spawners with short polyps, mixed types of tentacle shapes, and a phaceloid or flabello-meandroid skeleton. Soft-tissue morphology of *G. fascicularis* and *Ctenella chagius* were found to be consistent with the dichotomy.

**Conclusions**. The paraphyly of the original members of the previous subgenera justify recognizing *Fimbriaphyllia* as a genus. The integrated approach demonstrates that combining polyp features, reproductive traits, and skeletal morphology is of high systematic value not just to *Euphyllia* and *Fimbriaphyllia* but also to clade V; thus, laying the groundwork for resolving the phylogeny of clade V.

# INTRODUCTION

Systematics of the Scleractinia are traditionally based on features of the skeleton (also called the corallum) (*Dana, 1846*; *Edwards & Haime, 1857*; *Edwards & Haime, 1860*; *Vaughan & Wells, 1943*; *Veron & Pichon, 1980*; *Veron, 2000*). Despite the convenience of relying on the corallum, the skeleton is plagued with taxonomic ambiguities brought about by plasticity and convergence, which is a weakness in the traditional systematics of the Scleractinia. *Lang (1984)* proposed searching for other biological attributes for identification such as polyp or soft-tissue morphology and anatomy, mode of reproduction, behaviors, and ecological and physiological aspects of corals. These traits or a combination of any number of them are thought to have greater systematic value than adhering strictly to skeletal features (*Daly, Fautin & Cappola, 2003*). Recent advancements in molecular phylogenetic construction of the evolutionary history of scleractinian corals echo this proposal along with proposed major deviations from traditional classification schemes. The first deviation was the discovery of two major lineages within the order, now referred to as the robust and complex clades (*Romano & Palumbi, 1996*; *Romano & Palumbi, 1997*; *Romano & Cairns, 2000*; *Chen, Wallace & Wolstenholme, 2002*; *Fukami et al., 2008*; *Huang et al., 2009*; *Kitahara et al., 2010*; *Okubo, 2016*). The second deviation showed that paraphyly was found in 11 traditional families of the Scleractinia (*Fukami et al., 2008*; *Huang et al., 2009*; *Huang et al., 2011*; *Huang et al., 2014a*; *Huang et al., 2014b*; *Arrigoni et al., 2012*). These findings further suggested that skeletal features (colony formation, corallite diameter, and characteristics of the septa and costae) that are widely used in identifying species of corals are not fully reflective of evolutionary relationships within families and even between conspecific populations from the Atlantic, Pacific, and Indian Oceans (*Chen et al., 1995*; *Romano & Palumbi, 1996*; *Romano & Palumbi, 1997*; *Romano & Cairns, 2000*;

*Fukami et al., 2004*; *Fukami et al., 2008*; *Budd & Stolarski, 2009*; *Kitahara et al., 2010*; *Kerr, 2005*; *Arrigoni et al., 2012*; *Arrigoni et al., 2014a*; *Arrigoni et al., 2014b*; *Arrigoni et al., 2016*). These deviations have challenged systematists to reexamine phylogenetic groupings in contrast with the traditional families and to discern and propose characteristics, apart from the skeleton or other aspects of the skeleton, that are systematically informative and diagnostic of species in the new groupings. This integrated approach to systematics has led to remarkable resolutions in some scleractinian families. The Family Acroporidae, for example, one of the largest families in the Scleractinia, traditionally classified *Acropora* and *Isopora* as the two major subgenera in the family. Morphological characteristics and reproductive traits that were found to reflect the phylogenetic relationships in the family led to the recognition of *Acropora* and *Isopora* as independent genera (*Fukami, Omori & Hatta, 2000*; *Van Oppen et al., 2001*; *Wallace et al., 2007*). The characteristics and the attributes that were identified through the integrated approach have been demonstrated to be operationally useful in subsequent classifications in the group as in the example of *Isopora togianensis* (*Wallace et al., 2007*). Currently, the Mussidae, Merulinidae, Pectiniidae, and a new family, Lobophylliidae, are nearly completely phylogenetically resolved and revised (*Huang et al., 2009*; *Huang et al., 2011*; *Huang et al., 2014a*; *Huang et al., 2014b*; *Arrigoni et al., 2012*; *Arrigoni et al., 2014a*; *Huang et al., 2016*; *Budd et al., 2012*), but other groups, such as the family Euphylliidae (clade V), still lack their respective phylogenetic investigations.

The family Euphylliidae originally had 14 species classified into five genera namely the *Euphyllia, Cataliphyllia, Plerogyra, Physogyra*, and *Nemenzophyllia* (*Veron, 2000*). The *Euphyllia*, the largest genus in the family, was classified in earlier systematic schemes under the subfamily Eusmilinae of the Family Caryophylliidae (*Vaughan & Wells, 1943*; *Veron & Pichon, 1980*). *Veron & Pichon (1980)* recognized a dichotomy in the genus *Euphyllia* that is based on colony structure and represented this dichotomy as the subgenera *Euphyllia* and *Fimbriaphyllia*. The subgenus *Euphyllia* included species with a phaceloid growth form, which listed *E. glabrescens* and *E. cristata* in the group. The subgenus *Fimbriaphyllia*, on the other hand, included species with a flabello-meandroid growth form namely, *E. ancora* and *E. divisa*. The subgenera were eventually synonymized as *Euphyllia*; however, *Veron (2000)* retained the dichotomy based on colony structure as more species were discovered and classified under the genus *Euphyllia* (*Veron, 1990*). Eight species were then recognized and classified into two groups based on colony structure. One group, with phaceloid skeletons, included *E. glabrescens, E. cristata, E. paraancora, E. paradivisa*, and *E. paraglabrescens*; and the other group, with flabello-meandroid skeletons, included *E. yaeyamaenesis, E. divisa*, and *E. ancora* (*Veron, 2000*; Table 1). A ninth species, *E. baliensis*, with a phaceloid colony structure, was recently discovered from Bali, Indonesia (*Turak, Devantier & Erdmann, 2012*). Corallite features and tentacle shapes are both considered when classifying *Euphyllia* because dried skeletons that have the same colony structure are difficult to tell apart without viewing the live form (*Veron & Pichon, 1980*; *Veron, 2000*). Yet, an overlap in skeletal and tissue characteristics was also observed between species from different groups (Table 1). Recently, reproductive traits were recognized as an excellent guide for systematic affinities among the Scleractinia (*Kerr, Baird & Hughes, 2011*; *Baird, Guest & Willis, 2009*). For example, the phaceloid species, *E. glabrescens*, is hermaphroditic

with a reproductive mode of brooding, while the flabello-meandroid species, *E. ancora* and *E. divisa*, are gonochoric with a reproductive mode of broadcast spawning (Table 1; *Baird, Guest & Willis, 2009*).

Scleractinian phylogeny constructed using mitochondrial cytochrome oxidase I (*cox1*), cytochrome oxidase *b* (*cytb*), and nuclear β-tubulin showed that the family Euphylliidae is polyphyletic, with its members diverging from each other into the robust and complex clades (*Fukami et al., 2008*). *Physogyra lichtensteni* and *Plerogyra sinuosa* were clustered under clade XVI of the robust clade together with *Plesiastrea versipora* (*incertae sedis*) and *Blastomussa wellsi* (*incertae sedis*) (*Fukami et al., 2008*; *Arrigoni et al., 2012*; *Budd et al., 2012*; *Benzoni et al., 2014*). The genus *Euphyllia*, on the other hand, had some of its members cluster under clade V, which was still designated by *Fukami et al. (2008)* as the family Euphylliidae, of the complex clade. A paraphyletic pattern in the genus was already observed when *E. glabrescens* grouped with *Ctenella chagius* (Meandrinidae), and *E. ancora* and *E. divisa* grouped with *Galaxea fascicularis* (Oculinidae) (*Fukami et al., 2008*).

In this study, we utilized multiple genetic markers to construct the phylogeny of *Euphyllia* collected from the Philippines and Taiwan, including *E. paraancora, E. paradivisa, E. yaeyamaensis, E. divisa, E. ancora*, and *E. glabrescens*. The former three species do not have a clear phylogenetic status and/or have not yet been analyzed from the molecular perspective before. The phylogeny was used in examining the internal relationships in *Euphyllia* and in identifying the relevant morphological and reproductive traits to the systematics of the genus. The *Euphyllia* is currently the largest genus in clade V and the inferred traits from the internal phylogeny was extended to infer the external relationships of the genus to the other members of the clade; thus, laying the groundwork for the resolution of clade V.

## METHODOLOGY

### Sample collection and specimen identification

In total, 36 colonies representing six species of *Euphyllia* and *G. fascicularis* were collected and sampled from two areas of western Luzon, the Philippines and in Kenting National Park, Taiwan (Table S1). These two locations in the Philippines included Talim Bay, Lian, Batangas and Bolinao, Pangasinan; the latter is where *Veron (1990)* first found and described *E. paraancora*. All coral samples from the Philippines were collected through permissions granted by the Bureau of Fisheries and Aquatic Resources (BFAR) permit number FBP-0021-08. The sample from Taiwan was collected through permissions granted by the Kenting National Park Headquarters as part of a long-term monitoring program (Project 673202-LTER). All specimens were identified in the field using *Veron (1990), Veron (2000)*, and *Veron & Pichon (1980)*. Specimens were photographed underwater with a Canon A710 camera. After collection and tissue sampling, coral colonies were bleached, and skeletons were examined and kept at the Coral Museum of The Marine Science Institute, University of the Philippines.

The electronic version of this article in Portable Document Format (PDF) will represent a published work according to the International Commission on Zoological Nomenclature

**Table 1** **Characteristics of *Euphyllia*, *Galaxea*, and *Ctenella*.** Colony structure, corallites, tentacle morph, sexuality, and reproductive mode of *Euphyllia* spp, *Galaxea* sp., and *Ctenella chagius*. Data were modified from *Veron & Pichon (1980)*[a], *Veron (2000)*[b], *Sheppard, Dinesen & Drew (1983)*[c], and *Baird, Guest & Willis (2009)*[d]. Species names listed in Group 1 and Group 2 that are in bold are the original members of the subgenera *Euphyllia* and *Fimbriaphyllia* in groups 1 and 2 respectively.

| Species[a,b,c] | Colony structure[a,b] | Corallites[a,b,c] | Tentacle morph[b,c] | Sexuality[d] | Reproductive mode[d] |
|---|---|---|---|---|---|
| **Group 1** | | | | | |
| **E. glabrescens** | Phaceloid | First and second order septa plunge steeply near the centre of the corallite. Columella is absent. | Long tubular tentacles with knob-like tips | Hermaphroditic | Brooding |
| **E. cristata**[*] | Phaceloid | First and second order septa plunge steeply near the centre of the corallite. Columella is absent. | Long tubular tentacles with knob-like tips | Unrecorded | Unrecorded |
| E. paraglabrescens[*] | Phaceloid | Skeletons are almost identical to those of *E. glabrescens.* | Tentacles are short and bubble-like | Unrecorded | Unrecorded |
| E. paraancora | Phaceloid | Skeletons are like those of *E. glabrescens* with corallites 20–40 mm in diameter. | Tentacle tips form concentric circles and are shaped like an anchor, bean, or a kidney | Unrecorded | Unrecorded |
| E. paradivisa | Phaceloid | Skeletons are like those of *E. glabrescens.* | Branching tentacles almost identical to those of *E. divisa* | Unrecorded | Unrecorded |
| E. baliensis[*],[#] | Phaceloid | Corallites are sub-circular, with non-budding corallites averaging 3.1 mm diameter and ranging from 2–4.1 mm, with very thin walls | Tentacles are shaped like an anchor, kidney, or hammer at their tips, occasionally with additional smaller bulbous protuberances, the latter resembling mittens or gloves. | Unrecorded | Unrecorded |
| **Group 2** | | | | | |
| **E. divisa** | Flabello-meandroid | There are three orders of septa, which are exsert and plunge near the valley centre. Columella is absent. | Polyps have large tubular tentacles with smaller tubular branches. All branches have knob-like tips | Gonochronic | Broadcast spawning |
| **E. ancora** | Flabello-meandroid | Colonies have the same skeletal structure as *E. divisa* | Polyps have large tubular tentacles with few or no branchlets but with tips shaped like an anchor, bean, kidney, hammer, or a letter 'T' | Gonochronic | Broadcast spawning |

**Table 1** (*continued*)

| Species[a,b,c] | Colony structure[a,b] | Corallites[a,b,c] | Tentacle morph[b,c] | Sexuality[d] | Reproductive mode[d] |
|---|---|---|---|---|---|
| *E. yaeyamaensis* | Phaceloid/ Flabello- meandroid (with short valleys) | Septa occur in three orders and are usually compact. Columella is absent. | Tentacles are short and fleshy and covered with short uniform branch- lets, each with a termi- nal knob | Unrecorded | Unrecorded |
| **Group 3** | | | | | |
| *Galaxea fasicularis* | Plocoid | Corallites are of mixed sizes, usually less than 10 mm diameter with numerous septa reach- ing the corallite centre. Columella is absent. | Tubular tentacles with white tips, usually ex- tended during the day | Pseudogynodiecious | Broadcast spawning |
| **Group 4** | | | | | |
| *Ctenella chagius* | Meandroid | Two orders of septa with thick primary septa. Septa have small denticles and minute spinules. Columella is present. | Tubular tentacles ex- tended during the day | Unrecorded | Unrecorded |

**Notes.**

*not analyzed in this study.

#species described by *Turak, Devantier & Erdmann (2012)*.

(ICZN), and hence the new names contained in the electronic version are effectively published under that Code from the electronic edition alone. This published work and the nomenclatural acts it contains have been registered in ZooBank, the online registration system for the ICZN. The ZooBank LSIDs (Life Science Identifiers) can be resolved and the associated information viewed through any standard web browser by appending the LSID to the prefix http://zoobank.org/. The LSID for this publication is: urn:lsid:zoobank.org:pub:EBF69BA0-897E-4AC8-ADF5-4A7115CA1353. The online version of this work is archived and available from the following digital repositories: PeerJ, PubMed Central and CLOCKSS.

## DNA extraction and purification

A minimum of 1 cm$^3$ of a coral colony, includes both tissue and skeleton, was pruned off a sample with an orthopedic bone cutter. The pruned tissue with the skeleton was stored in pre-labeled 15 ml conical tubes containing CHAOS (Chaotropic solution: 4 M guanidine thiocyanate, 0.5% N-lauroyl sarcosine sodium salt, 25 mM Tris at pH 8, and 0.1 M 2-mercaptoethanol) (*Fukami et al., 2004*) that was 3∼5-times the volume of the sample taken (i.e., 1 cm$^3$ of tissue entailed 3∼5 mL of CHAOS). The tubes were kept in the dark for five days at room temperature. After the five-day period, DNA was obtained through the standard phenol/chloroform purification method with phenol extraction buffer (100 mM Tris-Cl at pH 8, 10 mM EDTA, and 0.1% sodium dodecylsulfate) (*Chen & Yu, 2000*; *Chen et al., 2000*). Purified DNA was quantified through spectrophotometry with Nanodrop 1000 by Thermo Fisher Scientific (Waltham, MA, USA) and through agarose gel electrophoresis (0.5% Seakem® LE agarose; Lonza, Basel, CH). The molecular weight

of each sample was estimated using the lambda ladder of Protech Technology Enterprise (Taipei, Taiwan).

## DNA sequencing

The *cytb* gene was amplified using newly developed primer pairs that were designed especially for *Euphyllia.* The *cytb* primer pairs have the following sequences: Eu4500F-1 (5′-CTG TCT AGT TTG GGA GTT AA-3′) and Eu4500R (5′-ATC ACT CAG GCT GAA TAT GC-3′) (set 1); and Eu4500F (5′-GAC AGA TGT TGT GCA ATG AG-3′) and Eu4500R-1 (5′-AAT AAG GCT ACC ATA AGC C-3′) (set 2). The expected product sizes of the amplicon for each pair were 1.0 and 1.5 kb, respectively. Two sets of primer pairs, developed by *Lin et al. (2011)*, that amplified the 3′-end region of scleractinian *cox1* were utilized: Cs-F17-a (5′-CCA TAA CCA TGC TTT TAA CGG ATA-3′) and Cs-R17-a (5′-TGC TAA TAC AAC TCC AGT CAA ACC-3′); and Cs-F18 (5′-GGA CAC AAG AGC ATA TTT TAC TG- 3′) and Cs-R18 (5′-CTA CTT ACG GAA TCT CGT TTG A-3′). The expected product sizes of the amplicon from each pair were 1,400 and 950 bp, respectively. The β-tubulin gene was amplified using a primer pair developed by *Fukami et al. (2004)*: forward (5′-GCA TGG GAA CGC TCC TTA TTT-3′) and reverse (5′-ACA TCT GTT GAG TGA GTT CTG-3′). The β-tubulin gene was expected to yield multiple gene copies with base pair lengths of 0.6, 1.5, and 2.0 kb (*Fukami et al., 2004*).

A polymerase chain reaction (PCR) was carried out in 50 μl reactions with either Pro-taq polymerase (Protech Technology Enterprise, Taipei, Taiwan) or Invitrogen[TM] taq-polymerase (Thermo Fisher Scientific, Waltham, MA, USA). Amplifications performed with the Invitrogen taq-polymerase contained a final concentration of the following: 2.0 mM of each base of dNTP, 3.0 mM of $MgCl_2$, 1× Invitrogen buffer (200 mM Tris-Cl at pH 8.4), 0.4 μM of primers, 1 unit of taq-polymerase, 2% DMSO, and at least 20 ng/μl of the DNA template. Amplifications performed with the Protech taq-polymerase, on the other hand, contained a final concentration of 2.0 mM of each base of dNTP, 1× Protech buffer (with $MgCl_2$), 0.4 μM of the primers, 5 units of the taq-polymerase, 2% DMSO, and at least 20 ng/μl of the DNA template. The PCR was carried out with a PxE Thermal Cycler by Thermo Fisher Scientific (Waltham, MA, USA). Amplification of the mitochondrial genes began with an initial denaturation temperature of 95 °C for 3 min, followed by 30 cycles of denaturation at 94 °C for 30 s, annealing at 50 °C for 1 min, and elongation at 72 °C for 90 s, with one final extension step at 72 °C for 10 min. The β-tubulin gene was amplified with an initial denaturation step at 94 °C for 2 min; followed by 30 cycles of denaturation at 94 °C for 45 s, annealing at 60 °C for 45 s, and elongation at 72 °C for 90 s, with one final extension step at 72 °C for 5 min. Among multiple copies of the β-tubulin gene of *Euphyllia*, the 600-bp band was selected for cloning because it was found to be present in all samples. Ligation and transformation of the β-tubulin amplicons was performed with a pGEM®-T Easy Vector System kit from Promega (Madison, WI, USA). Transformants were cultured in LB/ampicillin/IPTG/X-Gal plates, and five pure-white colonies were selected per sample. PCR products from all markers were purified with the PCR-M[TM] Clean-Up System of Viogene (New Taipei City, Taiwan) prior to sequencing.

## Phylogenetic analyses

Contiguous sequences (contigs) were assembled and annotated with Genious Pro vers. 4.6.1 software (*Drummond et al., 2011*). Each of the contigs was searched for in the database of NCBI BLAST to determine if the sequences matched a member of the Scleractinia. The search was also utilized to check for the direction of the sequences in the scleractinian sequences that they matched, especially with the clones of the β-tubulin gene. Sequences of a gene were then aligned using the CLUSTAL W plug-in of the software MEGA 5 (*Tamura et al., 2011*). *Cytb* and *cox1* segments from the complete mitochondrial genome of *E. ancora* (GenBank accession nos. NC015641 (*cytb*) and JF825139 (*cox1*); *Lin et al., 2011*) were used to guide the mitochondrial gene alignments. Available sequences of *E. glabrescens, E. ancora, E. divisa, G. fascicularis, C. chagius*, and other scleractinians were also gathered from the NCBI database.

   Phylogenetic trees were then generated for each set of genes and combined mitochondrial genes (*cox1* and *cytb*) using the Bayesian inference (BI) and maximum likelihood (ML) methods. The Bayesian inference was performed with the software Mr. Bayes 3.2.2 (*Ronquist et al., 2012*). Two runs were carried out with four Markov chains in 2 million generations, and the first 25% of trees were discarded as burn-in. Convergence of the BI analyses was determined by the average standard deviation (SD) of split frequencies (<0.01). ML trees were generated with 1,000 bootstrap replicates in MEGA versions 5 and 6 (*Tamura et al., 2011*). The best-fit models of evolution for the BI and ML analyses were obtained with jModeltest software (*Guindon & Gascuel, 2003*; *Posada, 2008*) (Table S2). The best-suited model was determined with a 95% confidence level using the Akaike information criterion (AIC) (*Posada & Buckley, 2004*).

## RESULTS

Both mitochondrial (*cytb* and *cox1*) and nuclear (β-tubulin) gene trees were congruent in terms of the general topologies, which showed strong statistical support for the clustering of all species of *Euphyllia* with other new members of the family Euphylliidae (*C. chagius* and *G. fascicularis*) (Figs. 1 and 2). Our mitochondrial and β-tubulin sequences were analyzed with the available sequences of *E. ancora* (JF825139 from *Lin et al., 2011*; AB441289 and AB441290 from *Fukami et al., 2008*), *E. divisa* (AB441288; *Fukami et al., 2008*), *E. glabrescens* (ABB441291, ABB441292, and ABB441377; *Fukami et al., 2008*), *G. fascicularis* (AB441286, AB441287, AB441374, AB441375; *Fukami et al., 2008*), and *C. chagius* (AB441378 and AB441379; *Fukami et al., 2008*) from clade V. From a wider context of scleractinian phylogeny, the gene trees of *Fukami et al. (2008)* consistently showed that clade VI and VII are the closest groups to clade V as they all share the same ancestor (Fig. S1). Sequences of *Pavona cactus* (AB441384 and AB441385 from *Fukami et al., 2008*), *Pavona clavus* (NC008165; *Medina et al., 2006*), and *Agaricia humilis* (AB441386 from *Fukami et al., 2008*; NC008160 from *Medina et al., 2006*) from clade VII were included to serve as the outgroup for our analyses (*Fukami et al., 2008*; *Lin et al., 2011*). A separate phylogenetic tree was generated with the mitochondrial sequences (combined *cox1* and *cytb*) of *Acropora tenuis* (AF338425, NC003522; *Van Oppen et al.,*

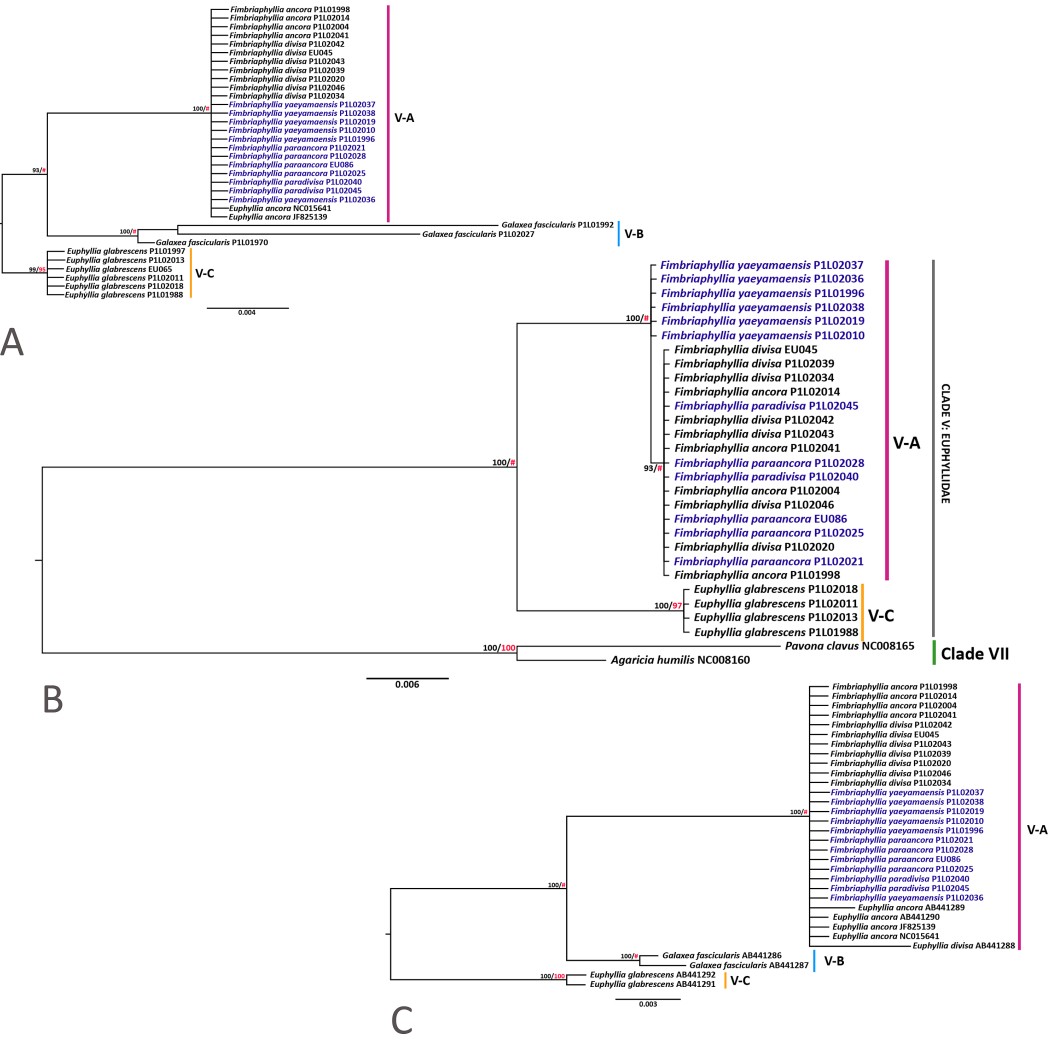

**Figure 1** **Phylogenetic trees of the *cox1*, *cytb*, and the combined *cox1* and *cytb* genes with sequences from clade VII as an outgroup.** The phylogenetic trees of the (A) *cox1* gene; (B) combined *cox1* and *cytb* genes; and (C) *cytb* gene. Bootstrap values of BI (black)/ML (red) are indicated before the nodes of the clusters. # indicates a difference in topologies between the BI and ML gene trees. Species names in blue font were analyzed herein for the first time. Distinct clusters in the tree are distinguished with vertical lines and labeled chronologically.

*2002*), *Anacropora matthai* (AY903295, NC006898; *Tseng, Wallace & Chen, 2005*), and *Montipora cactus* (AY903296, and NC006902; *Tseng, Wallace & Chen, 2005*) from clade VI (Fig. S2). The combined mitochondrial gene tree generated with clade VII as the outgroup produced the same topology as when clade VI is the outgroup (Fig. S2). Unfortunately, β-tubulin sequences of members of clade VI were not sequenced in the study of *Fukami et al. (2008)* and are not available at NCBI. Hence, gene trees generated with clade VII will be presented and referred to, mostly, for consistency.

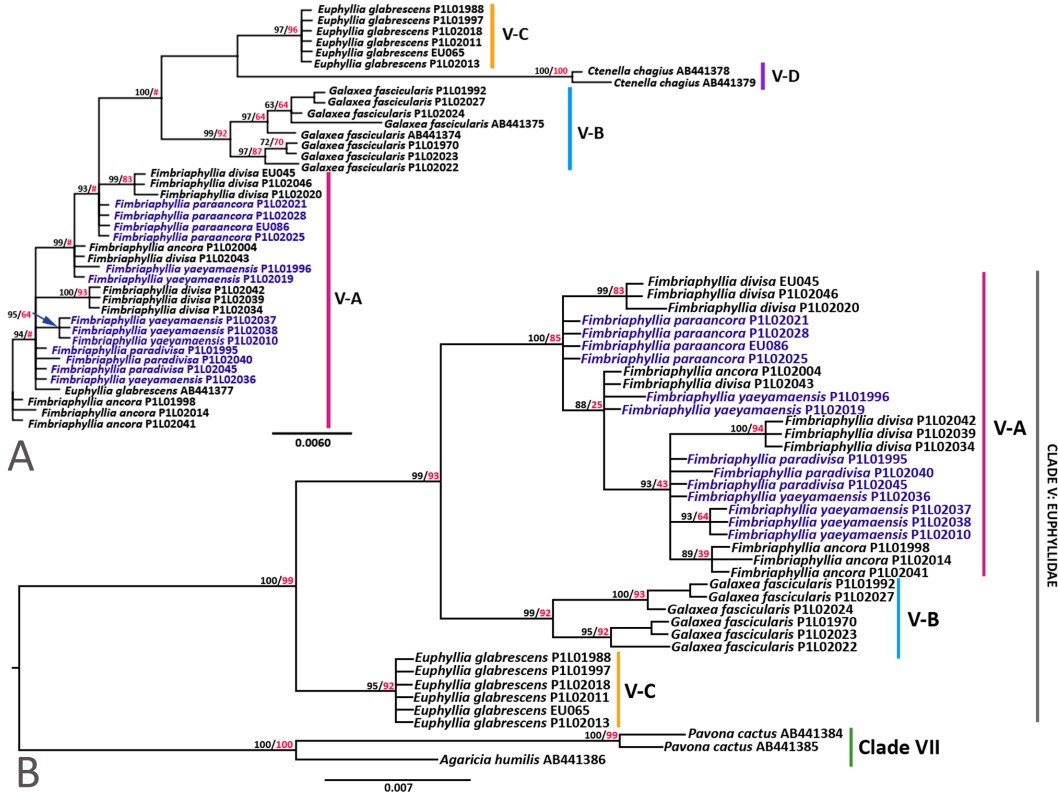

**Figure 2** **Phylogenetic trees of β-Tubulin.** The phylogenetic trees of β-tubulin gene showing (A) the un-rooted tree of *Euphyllia, Fimbriaphyllia, Galaxea,* and *Ctenella*; and (B) the gene tree of clade V with clade VII as the outgroup. Bootstrap values for BI (black)/ML (red) are indicated before the nodes of each cluster. # indicates a difference in topologies between the BI and ML gene trees. Species names in blue font were analyzed herein for the first time. Distinct clusters in the tree and the clades are distinguished with vertical lines and labeled accordingly.

A summary of the gene trees, gene lengths, and the model used to generate the tree is provided in Table S2. All our gene trees exhibited four major groups within the Euphylliidae with the available sequences (Figs. 1 and 2). *Euphyllia ancora, E. divisa, E. paraancora, E. paradivisa,* and *E. yaeyamaensis* were consistently clustered together (group V-A) and diverged from the same ancestor as *G. fascicularis* (group V-B), which formed a separate cluster of its own. All samples of *E. glabrescens* clustered together in group V-C, except for one instance where an *E. glabrescens* sequence (AB441377) clustered under group V-A in the β-tubulin gene tree (Fig. 2A). This may be accounted for by the multi-copy-nature of the β-tubulin gene. It is possible that among the many copies of β-tubulin, the sequence that is AB441377 is an ancient gene copy that may have similarities with the gene sequences of group V-A. Gene introgression poses another possibility; however, this may not be plausible because of the difference in mode of reproduction of *E. glabrescens* and members of group V-A, which will be discussed further later. The cluster of *E. glabrescens* mainly

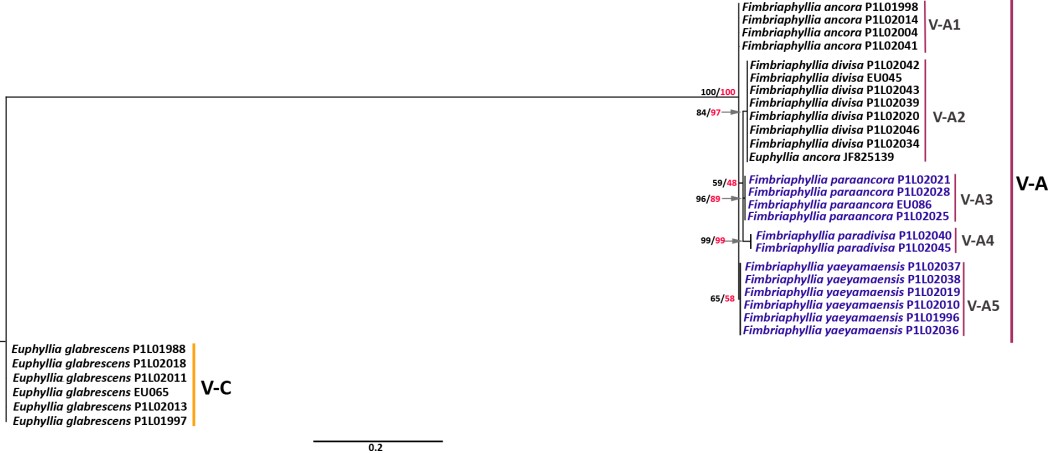

**Figure 3** **Phylogenetic tree of the 3′-end of the *cox1* gene.** Bootstrap values of BI (black)/ML (red) trees are indicated before the nodes of the clusters. Major groups in the tree are represented by V-A and V-C. Group V-A has five sub-groups (V-A1 to V-A5). All groups were distinguished with vertical lines and labeled chronologically. Species names in blue font were analyzed herein for the first time.

shares the same node with *C. chagius*, but the latter formed a group of its own that is highly divergent from *E. glabrescens* (group V-D).

Among the 22 complete scleractinian mitochondrial genomes examined by *Lin et al. (2011)*, an extra 699 bp at the 3′-end of the *cox1* gene was observed only in the whole mitochondrial genome sequence of *E. ancora* (NC015641). This extra region of the *cox1* gene was also found to occur in all samples of *Euphyllia* in the present study, but was not found in *G. fascicularis* and has not yet been reported in any other member of the Euphylliidae (clade V) as well as in other Scleractinia. Hence, a separate tree was generated for the 3′-end of the *cox1* gene to further examine the internal phylogeny within the genus (Fig. 3). The gene tree generated from the 3′-end of the *cox1* gene showed strong support for two general clusters, groups V-A and V-C, which is congruent with gene trees of β-tubulin, *cytb*, *cox1*, and the combined mitochondrial genes. More importantly, the gene tree demonstrated finer clustering with high supporting values for five distinct sub-clusters under group V-A. These sub-clusters were identified as *E. ancora* (V-A1), *E. divisa* (V-A2), *E. paraancora* (V-A3), *E. paradivisa* (V-A4), and *E. yaeyamaensis* (V-A5). It was, however, observed that one sample of *E. ancora* (JF825139) grouped in the sub-cluster of *E. divisa* (V-A2). *Euphyllia ancora* and *E. yaeyamaensis* each have their own distinct group. *Euphyllia divisa*, *E. paraancora*, and *E. paradivisa*, however, grouped separately from each other under one major cluster.

## DISCUSSION

### Recalling the subgenus *Fimbriaphyllia* and *Euphyllia*

*Veron & Pichon (1980)* classified four *Euphyllia* species into two subgenera under the subfamily Eusmilinae of the family Caryophylliidae. The two subgenera, namely

*Fimbriaphyllia* and *Euphyllia*, respectively represented a dichotomy based on colony structure. *E. ancora* and *E. divisa*, being flabello-meandroid in growth form were grouped separately from *E. glabrescens* and *E. cristata*, which had a phaceloid growth form (Fig. 4A, Table 1). Species within a subgenus were identified through polyp shapes as previous classification schemes did not report microstructure characteristics that distinguish between species (*Veron & Pichon, 1980*; Table 1, Fig. 4A). In all the gene trees, the *Euphyllia* has two distinct paraphyletic groups that are concordant with *Fukami et al. (2008)* and the phylogenetic tree from the 3′-end of the *cox1* of *Euphyllia* by *Lin et al. (2011)*. Group V-A represents the subgenus *Fimbriaphyllia* with the cluster of *E. ancora* and *E. divisa*, while group V-C represents the subgenus *Euphyllia* with a cluster of *E . glabrescens*. The paraphyly of the two groups with the original members of the subgenera retained in their respective clusters calls for a proposal to elevate the subgenus *Fimbriaphyllia* to genus. Consequentially, the proposal also calls for a revision of the previously established dichotomy between *Euphyllia* and *Fimbriaphyllia*, when they were still subgenera, especially since, based on phylogeny, *Fimbriaphyllia* gained new members. From here on, species in group V-A will bear the genus name *Fimbriaphyllia* when they are referred to in the succeeding discussion.

## Revising the dichotomy between *Euphyllia* and *Fimbriaphyllia*

While the subgenera were eventually synonymized as *Euphyllia*, the dichotomy based on colony structure was retained in the field guide of *Veron (2000)* as new species of *Euphyllia* were discovered and added to the genus (*Veron, 1990*). The present gene trees for *Euphyllia* still exhibit two major clusters as in the gene trees presented early on by *Fukami et al. (2008)* and *Lin et al. (2011)*. However, in contrast with the dichotomy of *Veron & Pichon (1980)* and *Veron (2000)*, the clustering of the phaceloid species of *Fimbriaphyllia paraancora* and *Fimbriaphyllia paradivisa* with the *Fimbriaphyllia* group effectively refutes the dichotomy based on colony structure. This finding is, in part, congruent with *Lin et al. (2011)*, where *F. paraancora* has already been observed to cluster with *E. ancora* and *E. divisa*. With the inclusion of *Euphyllia* species which were not analyzed before, the gene trees now support a dichotomy that is primarily determined by polyp shapes and polyp length instead (Fig. 4B). Furthermore, the phaceloid species, *F. paraancora* (group V-A), *F. paradivisa* (group V-A), and *E. glabrescens* (group V-C), were found to originate from different ancestral nodes in the family, suggesting that the phaceloid colony structure is a convergent trait. The clustering of *Fimbriaphyllia yaeyamaensis*, a species with both phaceloid and flabello-meandroid growth forms, with *Fimbriaphyllia* (group V-A) strengthens the dichotomy based on polyp morphology.

Members of *Fimbriaphyllia* are now characterized with polyps that have projections that are either anchor-shaped or branched (divided). These shapes are the bases of their species names. *Fimbriaphyllia ancora* and *F. paraancora* have anchor-shaped polyps, while *Fimbriaphyllia divisa, F. paradivisa*, and *F. yaeyamaensis* have branched (divided) polyps (Fig. 4B). Polyps in the *Fimbriaphyllia* group are also significantly shorter compared with polyps of the *Euphyllia* group. Polyp length, in this study, refers to the observed length of the polyps when they are fully inflated. This means that, polyps of corals from the

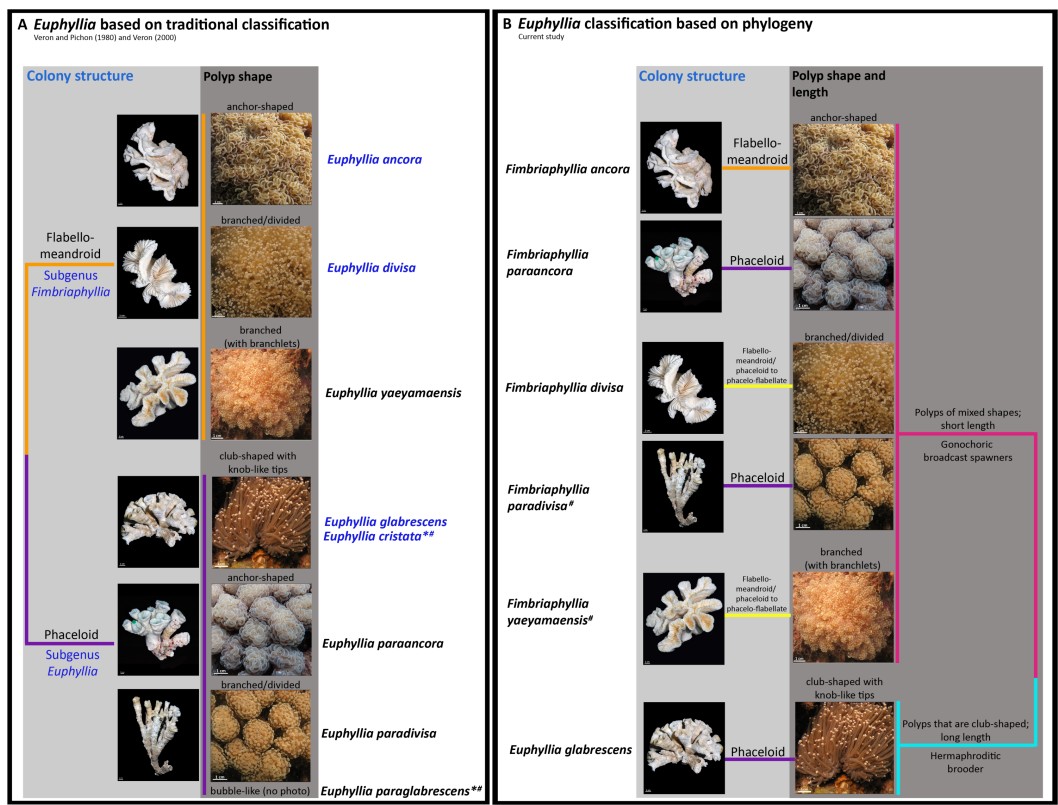

**Figure 4  Dichotomous trees of *Euphyllia* and *Fimbriaphyllia*.** (A) A dichotomous tree of *Euphyllia* based on *Veron & Pichon (1980)* and *Veron (2000)*. The species names in blue text are the original members of the subgenera *Euphyllia* and *Fimbriaphyllia* (*Veron & Pichon, 1980*). The dichotomy corresponds to the phaceloid and flabello-meandroid groups respectively, which *Veron (2000)* retained even with the synonymy of the subgenera to *Euphyllia* and as new species (in black font) were added to the genus. (B) The phylogenetic-based dichotomous tree, on the other hand, groups species according to polyp morphology and reproductive traits. Photographs of the polyps while fully inflated were taken carefully on field so as not to induce retraction. * species not analyzed in this study. # species with no known records of sexuality or reproductive mode.

*Fimbriaphyllia* group are still shorter than the polyps of the *Euphyllia* group even when they are fully inflated (Fig. 4B). Likewise, the length only refers to the regular polyps and not to sweeper tentacles that are, naturally, capable of extending to extreme lengths (i.e., as far as it can extend until it comes in contact with the coral/s beside it) for the purpose of defense. *Euphyllia glabrescens* (group V-C), on the other hand, has polyps that are aptly characterized as long, club-shaped, and glabrous because of the lack of protrusions or projections like branches or branchlets. Among species included in this study, this polyp shape was observed only in *E. glabrescens* and may be diagnostic for group V-C.

*Veron & Pichon (1980)* classified *E. cristata* under the subgenus *Euphyllia* and *Veron (2000)* classified *E. paraglabrescens*, together with *E. cristata* and *E. glabrescens*, with the phaceloid group. We were unable to obtain samples of *E. cristata*, *E. paraglabrescens*, and *E. baliensis* because of their rarity and limited range especially for *E. paraglabrescens* and

*E. baliensis*, which are known to occur only in Japan (*Veron, 1990*) and Indonesia (*Turak, Devantier & Erdmann, 2012*) respectively. *Euphyllia cristata* has the same polyp shape as *E. glabrescens*, but it is distinct from *E. glabrescens* in having relatively shorter polyps and an exsert primary septa, which can easily be observed when the polyps retract in the field. Given the polyp shape and polyp length of *E. cristata*, we predict from our inferences from the phylogeny presented here that *E. cristata* will cluster under group V-C. As the present study is undergoing review, our prediction has been confirmed with the recently published work of *Akmal et al. (2017)*, that also made use of the same universal primers from *Lin et al. (2011)*, which were also utilized in the present study. *Euphyllia paraglabrescens* and *E. baliensis*, on the other hand, are predicted to cluster with the *Fimbriaphyllia* group on account of having short polyps.

### Morphospecies in the *Euphyllia* and the *Fimbriaphyllia* and the systematic hierarchy

The phylogeny of the 3′-end of the *cox1* gene also supports two major groups (V-A and V-C) that represent the two genera and the dichotomy based on polyp morphology. The distinct clusters or subgroups in the gene tree were found to represent the morphospecies of *Euphyllia* and *Fimbriaphyllia*. In the sub-groupings of the *Fimbriaphyllia* group (sub-groups V-A1 to V-A5 in Fig. 3), species with flabello-meandroid colonies, except for *F. divisa*, clustered separately from each other as in *F. ancora* and *F. yaeyamaensis*. Species with a phaceloid colony structure, as in *F. paraancora* and *F. paradivisa*, each formed their own distinct group, but they were clustered together with the flabello-meandroid, *F. divisa*. In the major cluster of group V-A, *F. yaeyamaensis* and *F. ancora* are at the base of the group, which may mean that the phaceloid species in the group diverged later than the flabello-meandroid species. It appears that species with the same polyp shape have a phaceloid or a flabello-meandroid counterpart. For example, *F. ancora*, a flabello-meandroid species with anchor-shaped polyps, has *F. paraancora* as its phaceloid counterpart. *Fimbriaphyllia yaeyamaensis*, a species with branched polyps, has both flabello-meandroid and phaceloid colony structures. As suggested in the tree, the flabello-meandroid counterpart of *F. yaeyamaensis* with branched polyps may be *F. divisa*, and the phaceloid counterpart is *F. paradivisa*. However, *F. divisa* was also found to have both flabello-meandroid and phacelo-flabellate or phaceloid corallite structures. Our findings support the possibility that *Fimbriaphyllia* is still a young group and that introgression may still be occurring amongst its members; hence, reciprocal monophyly may not have been fully achieved yet. This phenomenon may also explain the grouping of *E. ancora* (JF825139) from Taiwan with *F. divisa* from the Philippines. Nevertheless, among the markers we used, the 3′-end of the *cox1* gene exhibited the best resolution as evidenced by the distinct clusters of species of the *Euphyllia* and the *Fimbriaphyllia* group in the gene tree. Apart from the capability to the resolve species phylogeny in *Euphyllia* and *Fimbriaphyllia*, as predicted by *Lin et al. (2011)*, the ability to tease-out sequences up to the species level and the uniqueness of the gene region to species of these genera shows potential for barcoding. So far, these characteristics are uncommon for the gene markers tested for Scleractinia; furthermore, it was established in the study of *Huang et al. (2008)* that the mitochondrial gene of anthozoans is slowly

evolving and is not suitable for the purposes of DNA barcoding. The 3′-end of the *cox1* gene of *Euphyllia* and *Fimbriaphyllia* may be an exception to the rule.

The phylogeny of the 3′-end of the *cox1* gene shows that it not only supports the dichotomy based on polyp morphology, but it also shows the combination of polyp and skeletal traits that are relevant in distinguishing species of *Fimbriaphyllia* and *Euphyllia* (Fig. 4B). As *E. glabrescens* is the only member of the *Euphyllia* group, so far as our study supports, tubular polyps and phaceloid colony structure is a combination of traits that have been found to be unique to the species. *Euphyllia cristata* shares the same combination of traits but its septal morphology has been described to be distinct from *E. glabrescens* (Veron & Pichon, 1980; Table 1). The finer clustering supporting the morphospecies and the distinction between *E. glabrescens* and *E. cristata* at the miscrostructure level opens the possibility that there may be microstructures that may also be able to diagnose species of *Fimbriaphyllia*.

## External relationships of the *Euphyllia* and *Fimbriaphyllia* with the new members of Euphylliidae (clade V)

Veron & Pichon (1980) and Veron (2000) described features of the Euphylliidae that we found to be shared features with *G. fascicularis* and *C. chagius*. The euphylliids, *G. fascicularis*, and *C. chagius*, are usually dome-shaped and massive, and yet coralla are light-weight with a blistery coenosteum. Colony structures of the Euphylliidae include phaceloid, flabello-meandroid, and meandroid. Septa are round or lobe-shaped with granulated or glabrous sides and margins. Walls are septothecal or parathecal. *Galaxea fascicularis* (Veron & Pichon, 1980) and *C. chagius* (Sheppard, Dinesen & Drew, 1983) also have granulated septa with variable costae that may be striated or ornamented with lobes or spines.

The clustering of *G. fascicularis* and *C. chagius* under the family Euphylliidae might not be surprising to some systematists. Previous classification schemes and species descriptions already suggested that *Galaxea* and *Ctenella* be grouped with *Euphyllia*. In the family tree of Scleractinia, Veron (2000) showed that the family Euphylliidae diverged from the family Oculinidae. In the same family tree, Oculinidae was shown to be grouped with the family Meandrinidae under the suborder Meandrina, where *C. chagius* was classified. Vaughan & Wells (1943) described *Euphyllia* and *Ctenella* together under the subfamily Eusmilinae of the Caryophylliidae (suborder Caryophyllida) on the bases of having exsert septa, intratentacular budding, septothecal walls, and a brown polyp color. Sheppard, Dinesen & Drew (1983) also suggested a revision of Meandrinidae stating that *Ctenella* be grouped under the Eusmilinae on the basis of having "smooth septal margins and a light-weight coralla". *Galaxea fascicularis* and *C. chagius* also have fleshy polyps that are usually extended during daytime. Sweeper tentacles and extracoelomic digestion were also documented in *Euphyllia, Galaxea*, and *C. chagius* (Sheppard, Dinesen & Drew, 1983; Hidaka, 1985; Borneman, 2001). While *C. chagius* is meandroid, which is already a known trait of the Euphylliidae, the inclusion of *Galaxea* adds a plocoid growth form to the family (clade V).

The affinity of *G. fascicularis* and *C. chagius* with euphyllids is, so far, consistent with the dichotomy based on polyp morphology. *Galaxea fascicularis* has short polyps as in species of *Fimbriaphyllia*, while *C. chagius* has the same long and tubular polyps as in *E. glabrescens.* Despite the shared characteristics of *C. chagius* with the Euphylliidae, *C. chagius* is highly divergent in the Euphylliidae (clade V). This divergence may indicate differences in the skeletal morphology of *C. chagius* with *Euphyllia* and *Fimbriaphyllia*. *Galaxea fascicularis, Euphyllia*, and *Fimbriaphyllia* species usually have three to four or sometimes five orders of septa and the columella is often weakly developed or absent (*Veron & Pichon, 1980*; *Veron, 1990*; *Veron, 2000*). *Ctenella chagius*, on the other hand, is characterized by two orders of septa and the presence of a lamellar or continuous columella (*Sheppard, Dinesen & Drew, 1983*). The high divergence may also be attributed to the limited geographical range of *C. chagius* as it has only been reported to occur locally in the Chagos Archipelago of the Indian Ocean (*Sheppard, Dinesen & Drew, 1983*). There have been no new records of the species elsewhere to date (*Sheppard, Turak & Wood, 2008*; *Carpenter et al., 2008*). In relation to having restricted species ranges, the systematic positions of *Gyrosmilia* and *Montigyra*, monotypic genera of the family Meandrinidae that are found only in the Indian Ocean, have not yet been analyzed from the molecular perspective. All our samples of *Euphyllia, Fimbriaphyllia*, and *Galaxea* are from the Pacific Ocean, but their geographic ranges extend until the Indian Ocean (*Veron, 2000*). Hence, the inclusion of samples of euphylliids from the Indian Ocean may strengthen the phylogeny presented here. On the other hand, there is a possibility that Indian Ocean specimens may also show divergence between populations of the same species collected from the Pacific and Indian Oceans (*Fukami et al., 2004*) as in *Favites complanata, Dipsastraea rotumana*, and *D. pallida* (clade XVII) (*Arrigoni et al., 2012*; *Budd et al., 2012*).

## Evolution of reproductive traits in the Euphylliidae (clade V)

Sexuality and modes of reproduction are emergent patterns that were also perceived in the clustering in the gene trees. *Fimbriaphyllia ancora*, *F. divisa*, and *F. paraancora* (group V-A), are mainly gonochoric (dioecious) broadcast spawners (*Borneman, 2003*; *Twan, Hwang & Chang, 2003*; *Twan et al., 2005*; *Twan et al., 2006*; *Fan et al., 2006*). To date, there is no scientifically published information about the reproductive mode of *F. paradivisa* and *F. yaeyamaensis*, but we predict that they are also gonochoric broadcast-spawners as in other species of *Fimbriaphyllia*. *Euphyllia glabrescens* (group V-C), on the other hand, is known to be a hermaphroditic brooder (*Richmond & Hunter, 1990*; *Baird, Guest & Willis, 2009*). Sexuality and reproductive modes are still unknown for *E. cristata*, *E. paraglabrescens*, and *E. baliensis. Galaxea fascicularis* was first reported to be a hermaphroditic broadcast-spawner (*Babcock et al., 1986*; *Shlesinger, Goulet & Loya, 1998*) but was later reported to be a pseudo-gynodioecious broadcast-spawner instead (*Guest et al., 2005*; *Baird, Guest & Willis, 2009*). *Keshavmurthy et al. (2012)* were able to resolve the sexuality of the species by showing that *G. fascicularis* is gynodioecious rather than pseudo-gynodioecious. Being gynodioecious is characterized by the existence of female colonies separate from hermaphroditic colonies that produce viable egg and sperm (*Keshavmurthy et al., 2012*). Pseudo-gynodioecious, on the other hand, is essentially the same, but hermaphroditic

colonies are thought to produce non-viable eggs, hence "pseudo" (*Guest et al., 2005*; *Baird, Guest & Willis, 2009*). The gynodioecious type of sexuality has not yet been documented in other corals and is, to date, unique to *G. fascicularis* (*Keshavmurthy et al., 2012*). The sexuality of *G. fascicularis*, being gynodioecious, appears to be an intermediate or a "transitional state" between *Fimbriaphyllia* (dioecious) and *Euphyllia* (hermaphroditic), which was also mentioned by *Keshavmurthy et al. (2012)*. This claim is supported by our gene trees, and it may explain why *G. fascicularis* is in its own cluster, but its close affinity to *Fimbriaphyllia* is based on being a broadcast-spawner. Unfortunately, the sexuality and mode of reproduction of *C. chagius* has not yet been documented, but as the pattern in our gene tree suggests, *C. chagius* is hypothesized to be a hermaphroditic brooder like *E. glabrescens*.

The gene trees of *Fukami et al. (2008)* clustered *Euphyllia, Galaxea*, and *Ctenella*, which are now new members of the family Euphylliidae (clade V; *Fukami et al., 2008*). The general topology in our gene trees is concordant with the gene trees of *Fukami et al. (2008)* and *Lin et al. (2011)* even with the inclusion of *F. yaeyamaensis, F. paraancora*, and *F. paradivisa*. The inclusion of other species of *Euphyllia* in the phylogenetic reconstruction of the Euphylliidae led to the recognition of the former subgenera in the *Euphyllia*, which called for a proposal to recall and elevate *Fimbriaphyllia* as a genus. The phylogeny of *Euphyllia* and the *Fimbriaphyllia* shows that systematics need not be limited to skeletal traits and the distinction between morphospecies highlights the importance of combining polyp morphology and reproductive traits with skeletal morphology, which are of higher systematic value.

## Systematic account

**Family Euphylliidae** *Alloiteau, 1952*

Euphylliidae *Alloiteau, 1952*
Euphyllidae *Veron, 2000*

Species of the family Euphylliidae have coralla that are massive, yet light-weight, with corallites that have thin walls and meandroid, plocoid, flabello-meandroid, or phaceloid formations. The septa are granulated and have shapes that vary between lobed, square, or angular. There are three to five orders of septa depending on the species. The first order septa are usually exsert. Columellae are absent in the calices of *Euphyllia* and *Fimbriaphyllia*. The costae occur as fine striations or as a series of thickened lobes that may also have spines. The series of thickened lobes may form rings around phaceloid corallites. These may indicate the previous height of the branch. These thickenings are also found in flabello-meandroid colonies. Walls are septothecal or parathecal. The coenosteum is generally smooth but it may also be blistery and may be lined with longitudinal ridges.

Species of the family Euphylliidae are well-known for their fleshy and colourful polyps, which makes them easily recognizable in the field. Tentacles have varied shapes, which may be long and glabrous or short with projections that may be branched or anchor-shaped. Sweeper tentacles have been observed in aquaria and in the field in a number of species.

Genus *Catalaphyllia* *Wells, 1971*
Genus *Ctenella* *Matthai, 1928*
Genus *Euphyllia* *Dana, 1846*
Genus *Fimbriaphyllia* *Veron & Pichon, 1980*
Genus *Galaxea* *Oken, 1815*
Genus *Gyrosmilia* *Milne Edwards & Haime, 1851*
Genus *Montigyra* *Matthai, 1928*
Genus *Simplastrea* *Umbgrove, 1939*

**Genus  *Euphyllia*** *Dana, 1846*

*Euphyllia* *Dana, 1846*; *Matthai, 1928*; *Veron, 2000*
*Leptosmilia* *Milne Edwards & Haime, 1848*
*Botryphyllia* *Shirai, 1980*
*Euphyllia* (*Euphyllia*) *Veron & Pichon, 1980*

**Characters.** Species of *Euphyllia* are characterized by phaceloid corallite formations. Septa are granulated, and may have a lobed or angular shape. The septa are in four to five orders and the first order septa are typically exsert. Septal margins are generally smooth but have areas with fine serrations when examined up-close. Costae vary from fine striations to small and short thickened lobes, sometimes, with spines. Polyps are fleshy, long, and glabrous with knob-like tips that are extended during the day.
**Type species:** *Euphyllia glabrescens* (*Chamisso & Eysenhardt, 1821*)

***Euphyllia glabrescens***  (*Chamisso & Eysenhardt, 1821*)
  (Figs. 5A–5E)

*Caryophyllia glabrescens* *Chamisso & Eysenhardt, 1821*
*Caryophyllia angulosa* *Quoy & Gaimard, 1824*
*Euphyllia glabrescens* *Nemenzo, 1960*; *Veron & Pichon, 1980*; *Veron, 2000*
*Euphyllia* (*Euphyllia*) *glabrescens* (*Chamisso & Eysenhardt, 1821*)
*Euphyllia costulata* (*Milne Edwards & Haime, 1849*)
*Euphyllia gaimardi* (*Milne Edwards & Haime, 1849*)
*Euphyllia laxa* *Gravier, 1910*
*Euphyllia rugosa* *Dana, 1846*
*Euphyllia striata* (*Milne Edwards & Haime, 1849*)
*Euphyllia turgida* *Dana, 1846*
*Leptosmilia costulata* *Milne Edwards & Haime, 1849*
*Leptosmilia gaimardi* *Milne Edwards & Haime, 1849*
*Leptosmilia glabrescens* (*Chamisso & Eysenhardt, 1821*)
*Leptosmilia rugosa* *Dana, 1846*
*Leptosmilia striata* *Milne Edwards & Haime, 1849*
*Lobophyllia glabrescens* (*Chamisso & Eysenhardt, 1821*)

**Material studied:** (P1L01988), Clubhouse, Talim Bay, Lian, Batangas, Philippines, coll. Dec. 2007, <5.0 m (Figs. 5A–5D); (P1L02018), Clubhouse, Talim Bay, Lian, Batangas,

Philippines, coll. Apr. 2008, <5.0 m (Fig. 5E); (P1L02011, P1L02017), Clubhouse, Talim Bay, Lian, Batangas, Philippines, coll. Apr. 2008 and Jul. 2008, <5.0 m.

**Description.** Colonies have phaceloid corallite formation with thin walls (Fig. 5A). Calice diameters range from 15.80 mm to 38.25 mm and branch diameters range from 12.90 mm to 27.35 mm (Fig. 5B). Branches have heights that range from 41.25 mm to 51.45 mm from the main point of branching. Septa occur in four to five orders. They generally have a lobed, sometimes angular, shape. For example, the first order septa may form a ridge first that is approximately 80 to 90° to the edge of the wall (Fig. 5C). The ridge will then turn down sharply as it reaches the center of the corallite and then plunge steeply giving the septa a square or angular shape when the corallite is viewed at eye-level. Sides of the septa are granulated. The septal margins are generally smooth but have areas with fine serrations when examined up-close. Columella are absent. Costae are in the form of fine striations that make the outer wall look smooth (Fig. 5D). Costae of this kind are observed especially for the corallites located at the periphery of the colony and those that have long branches. Corallites in the inner part of the colony and those that are in the process of budding or have just successfully budded exhibit costae in the form of small, thickened lobes that may have spines (Fig. 5C). A series of adjacent lobes may form a thickened ring around the periphery of the corallite (Fig. 5C). As the corallite increases in height, the ring remains and the corallite builds a coenosteum and a new ring of costae forms above the previous ring. These thickened rings may be observed throughout the length of the branch and may indicate the previous height of the branch. The coenosteum is generally smooth with a few longitudinal ridges that lightly run across the branches (Fig. 5E).

Live colonies have polyps that are fleshy, long, and glabrous with knob-like tips that are fully extended during the day. Observed polyp colors include brown and bright, fluorescent green. The knob-like tips are white in color. Colonies possess sweeper tentacles for defense. Colonies have a limited distribution range in the reef. They are found on the shallow parts of the reef slope covering the depths of around 3.0 m to 6.5 m. The species has also been observed to thrive in silty areas or reefs with turbid waters.

**Comparisons with related taxa.** The coralla of *E. glabrescens* are similar to those of the phaceloid species of *Fimbriaphyllia* but differ in calice diameter, branch height, and costae type. Colonies of *Fimbriaphyllia* have taller colonies compared with *E. glabrescens*. Calices of *Fimbriaphyllia paraancora* are larger than *E. glabrescens*, while calices of *Fimbriaphyllia paradivisa* are smaller than *E. glabrescens*. Polyps of *E. glabrescens* are significantly longer than the fully extended polyps of *Fimbriaphyllia* species. *E. glabrescens* has glabrous polyps, while *Fimbriaphyllia* species have mix types of polyp shapes (i.e., branched, anchor-shaped).

Living *E. glabrescens*, especially with fully extended polyps, may also be mistaken for *Heliofungia actiniformis*. The polyps of both species are long and glabrous, with knob-like tips. *H. actiniformis*, however, is solitary and free-living with bigger polyps that exhibit dark meandering lines that radiate from the center of the colony to the edge of the coral.

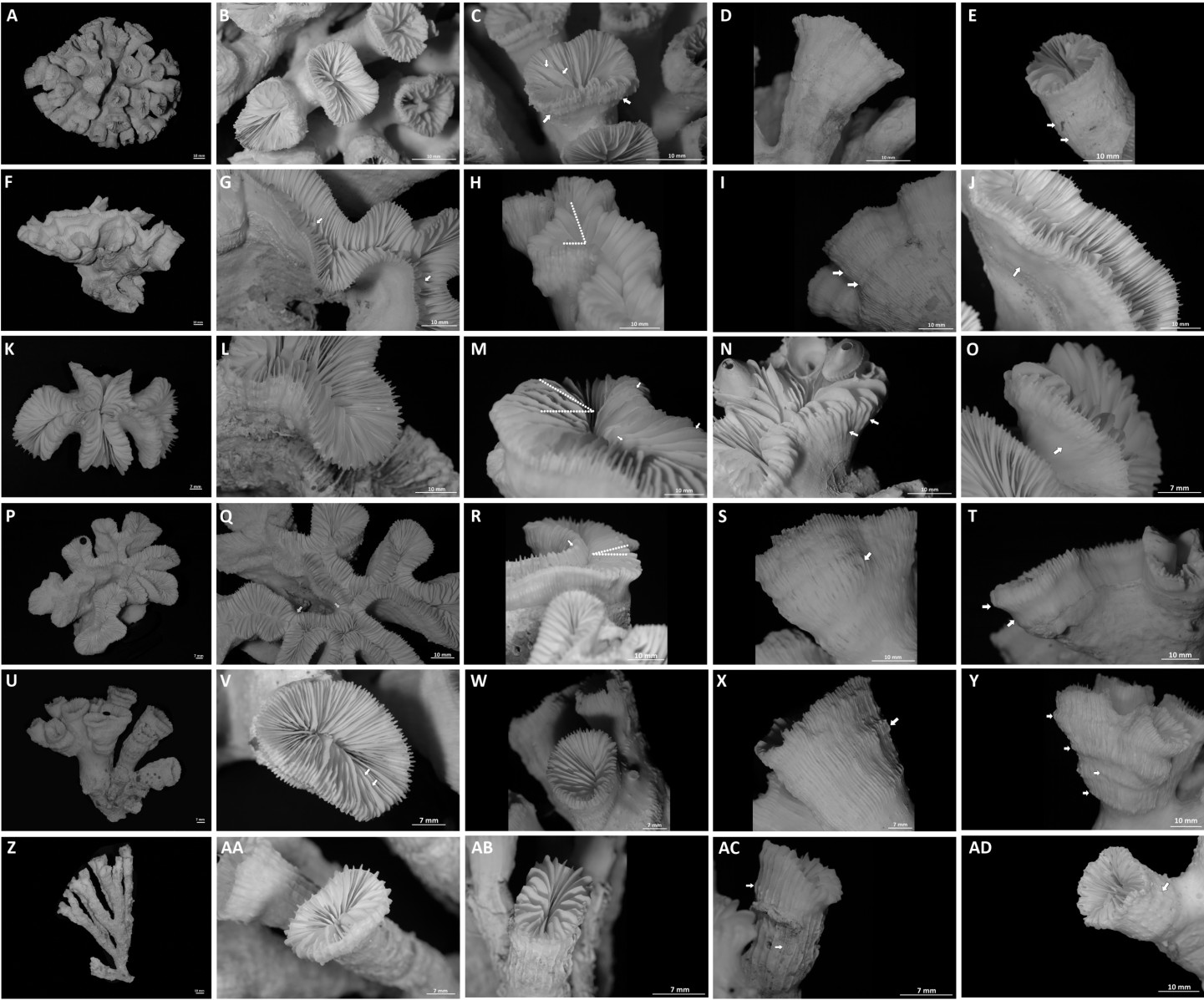

**Figure 5** **Skeletal morphology of *Euphyllia* and *Fimbriaphyllia* species.** *Euphyllia glabrescens* exhibit (A) phaceloid corallite formation with (B) round to oval calices bearing (C) four to five orders of septa. Septa may be lobed or angular (upper arrows). (D) Costae are fine striations and the (E) wall may have (arrows) ridges found throughout the length of the branch. *Fimbriaphyllia ancora* exhibit (F) flabello-meandroid corallite formation with (G) valleys bearing two to three orders of septa. Arrows show curling of the septa along the center of the valley. (H) Septa are steep and lobe-shaped. (I) Arrows point to costae, which may be ridges or bumps that form along the meandering parts of the wall. (J) Fine striations (arrows), on the other hand, form along the walls of straight valleys giving the wall a smooth appearance *Fimbriaphyllia divisa* exhibit phaceloid or phacelo-flabellate and (K) flabello-meandroid colonies with (L) valleys bearing three to four orders of septa. (M) Septal angles are less steep compared with *F. ancora* and may exhibit (upper arrow) the lobe shape and (two lower arrows) the angular shape. (N) The arrows point to the costae, which are lobe or blade-like protuberances that are concentrated on the upper part of the wall. (O) In some specimens, the wall may be smooth. *Fimbriaphyllia yaeyamaensis* exhibit both phaceloid and (P) flabello-meandroid colonies with (Q) valleys having three to four orders of septa. Arrows point to walls that form across valleys. (continued on next page...)

**Figure 5 (…continued)**
(R) Septal angles are low especially on the flaring side of the meandering valley, while angles are steep on the opposite side (arrow on the left). (S) Arrow points to costae in the form of long prominent ridges. (T) Some specimens have a concentration of the costae on the upper part, near the corallite, and the rest of the wall is smooth. *Fimbriaphyllia paraancora* exhibit the (U) phaceloid corallite formation. (V) Calices are wide with round to oval shape bearing four orders of septa. Arrows point to the wavy curves along the septal slope. (W) Septa steeply curve down to the center of the corallite making the calice deep. (X) Costae are in the form of ridges or lobes. The latter may be in series forming a thickened ring (arrow) near the calice. (Y) Arrows point to thickenings along the branch, which may indicate the previous height of the branch. *Fimbriaphyllia paradivisa* exhibit the (Z) phaceloid corallite formation. (AA) Calices are small bearing three orders of septa. (AB) Septa are generally lobe-shaped but some may have an angular shape. (AC) Costae are in the form of ridges or (AD) small lobes, which are widely-spaced from each other along the branch.

**Occurrence:** Philippines (*Veron & Hodgson, 1989*; this publication); India (*Pillai, 1971*); Taiwan (*Chen et al., 2005*); Malaysia (*Mazlan et al., 2005*; *Khodzori et al., 2015*; *Akmal et al., 2017*; Australia (*Griffith, 2004*), Japan (*Veron, 1990*).

**Genus** *Fimbriaphyllia* *Veron & Pichon, 1980*

*Euphyllia* *Dana, 1846* (in part); *Veron, 2000*
*Euphyllia (Fimbriaphyllia)* *Veron & Pichon, 1980*

**Characters.** Coralla of the genus have phaceloid or flabello-meandroid corallite formations. The septa are granulated, typically exsert, and have three or four orders. The shape of the septa is generally lobe-shaped and/or angular, depending on the species. The margins appear to be smooth but may have fine serrations when examined up-close. Columella are absent. Costae may be in the form of fine striations or prominent ridges that extend down the wall from the rim of the corallite. The coenosteum may be smooth or blistery. Ridges, that may be prominent or light, which may, in some species, run down the length of the wall.

Polyps are fleshy and are fully extended during day time but easily retract upon contact. They are short relative to the polyps of *Euphyllia* species, even when fully extended. The polyps of *Fimbriaphyllia* have a variety of shapes, which include anchor, kidney, bean, and branched. Sweeper tentacles have been observed in a number of species in the group. Species are widely distributed along the reef slope.

**Taxonomic notes.** *Veron & Pichon (1980)* introduced the subgenera *Fimbriaphyllia* and *Euphyllia* under the genus *Euphyllia*. They did not designate a type for the new taxon *Euphyllia (Fimbriaphyllia)*, but classified two new species under that subgenus, *Euphyllia (Fimbriaphyllia) ancora* *Veron & Pichon (1980)* and *Euphyllia (Fimbriaphyllia) divisa* *Veron & Pichon (1980)*. They also mentioned *Euphyllia fimbriata* (*Spengler, 1799*) as a candidate species for the new subgenus because of its skeleton morphology, but they considered its name a *nomen dubium* because the structure of the polyps was not recorded and because the type locality was unknown. These are not sufficient reasons to declare a species name to be a *nomen dubium* as long as the type is not lost, but the whereabouts of the type was not discussed. According to *Wells (1971)* one of the types of *Euphyllia plicata Milne Edwards & Haime, 1848* was illustrated by *Matthai (1928)* as *Euphyllia fimbriata*. Both species names were placed in the synonymy of *Catalaphyllia plicata* (*Milne Edwards & Haime, 1849*) by *Wells (1971)*. According to *Veron & Pichon (1980)* this was not correct

and they considered the specimen illustrated by *Matthai (1928)* as a representative of the *Euphyllia (Fimbriaphyllia)* group and they redesignated *Pectinia jardinei Saville-Kent, 1893* as the type species of *Cataluphyllia Wells, 1971*.

   The two subgenera, *Euphyllia* and *Fimbriaphyllia*, were synonymized by *Veron (2000)* as *Euphyllia*. Molecular analyses show that *Euphyllia ancora* and *Euphyllia divisa*, those that were originally classified by *Veron & Pichon (1980)* under the subgenus *Euphyllia (Fimbriaphyllia)* cluster together. This suggests that *Fimbriaphyllia* may be recognized as a genus separate from *Euphyllia,* which now necessitates a revision and the designation of a type species for the new genus.

**Type species:** *Fimbriaphyllia ancora* (*Veron & Pichon, 1980*). Designated herein.

***Fimbriaphyllia ancora*** (*Veron & Pichon, 1980*)
   (Figs. 5F–5J)

   *Euphyllia (Fimbriaphyllia) ancora Veron & Pichon, 1980*
   *Euphyllia ancora Veron, 2000*

**Material studied:** (P1L02170), Reyna, Talim Bay, Lian, Batangas, Philippines, coll. Apr. 2010, 4.0 m (Figs. 5F, 5G, 5I); (P1L02184), Outer Talim, Talim Bay, Lian, Batangas, Philippines, coll. Apr. 2010, 10.0m (Figs. 5H, 5J); (P1L01998), Talim Inner, Talim Bay, Lian, Batangas, Philippines, coll. Jan. 2008, <5.0 m; (P1L02164, P1L02167), Reyna, Talim Bay, Lian, Batangas, Philippines, coll. Apr. 2010, 6.0 m; (P1L02170), Layag-Layag, Talim Bay, Lian, Batangas, Philippines, coll. Apr. 2010, 4.0 m.

**Museum Repository:** Coral Museum, The Marine Science Institute, University of the Philippines

**Description.** Corallite formation is flabello-meandroid (Fig. 5F) with valley widths that range from 7.45 mm to 19.00 mm in larger colonies and 5.70 mm to 15.40 mm in smaller colonies. The length of the valley ranges from 34.50 mm to 100.80 mm from the common point of divergence with other valleys among the larger specimens. Septa are granulated and generally lobe-shaped. There are two to three orders of septa with the lower orders arranged in between the first order septa (Fig. 5G). The first order septa are typically exsert. The width of the septa from the wall to the valley center ranges from 4.90 mm to 14.30 mm in larger specimens and 4.30 mm to 7.25 mm in a smaller colony. Septa, especially along the straight valleys of the corallum, have steep angles ranging from 37 º to 64 º from the horizontal plane of the valley (Fig. 5H). Septa that are found along the meandering parts of the corallum may have lower angles. The septa plunge steeply towards the center of the valley with its edge curling parallel to the valley (Fig. 5G). Columella are absent. The outer wall varies from being rough (Fig. 5I) or smooth depending on the form of the costae (Fig. 5J). The costae may be in the form of longitudinal ridges and round bumps (Fig. 5I) or fine striations (Fig. 5J). The ridges are perpendicular to the upper edge of the wall and may extend down the height of the wall. Fine striations form along the wall of non-meandering valleys while the ridges form along the wall of the meandering parts.

   Live colonies have fully extended polyps during the day. They exhibit short and fleshy polyps that take on the shape of anchor, kidney, or bean. The polyps are typically dark

green in color with light-green tips but variations of brown stalks with white tips, or pink stalks with light pink tips have also been encountered on field. A branching variant of the anchor-shaped polyps has also been observed *in situ* but is rare. The species also possess sweeper tentacles.

*F. ancora* colonies are distributed all over the reef from the reef edge, with a depth of 60m, to the shallow parts of the reef crest with a depth of 5m or less. Colonies that inhabit the deeper parts of the reef were found in silty bottoms, while those in the reef slopes may sometimes be found sheltered in concavities. Colonies found in the silty parts of the reef are not attached to the substrate and can easily be collected.

**Comparison with related taxa.** *F. ancora* colonies have the same flabello-meandroid formation as *Fimbriaphyllia divisa* and *Fimbriaphyllia yaeyamaensis*. *F. ancora* has been found to have round or oval-shaped septa that are similar to *F. divisa.* However, *F. ancora* has significantly steeper septal slopes compared with *F. divisa. F. yaeyamaensis* has less steep slopes with broad, square-shaped septa as opposed to the steep, round or oval-shape septa that is characteristic of *F. ancora. F. ancora* has anchor-shaped polyps while *F. divisa* has branched polyps and *F. yayamaensis* has branched polyps with branchlets.

*F. ancora* shares similarities with *Catalaphyllia* and *Plerogyra sinuosa. Catalaphyllia* has wider V-shaped valleys compared with *F. ancora.* Septa of *Catalaphyllia* gradually plunge into the center of the valley from the wall, while the septa of *F. ancora* would flare into a lobe shape first before plunging steeply into the center of the valley. Septa of *F. ancora* are also more exsert than *Catalaphyllia. P. sinuosa,* septa are significantly more exsert than those of *F. ancora.* The latter's septa are also widely spaced and not as ordered as in *P. sinuosa.*

**Taxonomic notes.** The two original syntypes of *Euphyllia (Fimbriaphyllia)* are *Euphyllia ancora* and *E. divisa.* The designation of *E. ancora* as the type species of the genus *Fimbriaphyllia* is based on its polyps having an older historical record than *E. divisa.* *Veron & Pichon (1980)* noted under the description of *E. ancora* that the species has been illustrated by *Saville-Kent (1893)* as *Rhipidogyra.* This is probably the oldest illustration of the species. In addition, although both *E. ancora* and *E. divisa* are common within their respective distribution ranges, *E. ancora* is more widespread than *E. divisa* (*Veron, 2000*).

**Occurrence:** Philippines (*Veron & Hodgson, 1989*; this publication), Taiwan (*Chen et al., 2005*); Malaysia (*Mazlan et al., 2005*; *Khodzori et al., 2015*; *Akmal et al., 2017*), Australia (*Griffith, 2004*; *Richards & Beger, 2013*), Japan (*Veron, 1992*)

***Fimbriaphyllia divisa*** (*Veron & Pichon, 1980*)
(Figs. 5K–5O)

*Euphyllia (Fimbriaphyllia) divisa* *Veron & Pichon, 1980*
*Euphyllia divisa* *Veron, 2000*

**Material studied:** (P1L02172), Reyna, Talim Bay, Lian, Batangas, Philippines, coll. Apr. 2010, 7.9 m (Figs. 5K, 5O); (P1L02043) Clubhouse, Talim Bay, Lian, Batangas, Philippines, coll. Apr. 2008, <5.0 m (Fig. 5L); (P1L02034), Clubhouse, Talim Bay, Lian, Batangas,

Philippines, coll. Apr. 2008, <5.0 m (Fig. 5M); (P1L02042) Clubhouse, Talim Bay, Lian, Batangas, Philippines, coll. Dec. 2007, <5.0 m (Fig. 5N); (P1L02177), Reyna, Talim Bay, Lian, Batangas, Philippines, coll. Apr. 2010, 7.4 m; (P1L02181), Outer Talim, Lian, Batangas, Philippines, coll. Apr. 2010, 10.5 m; (P1L02183), Outer Talim, Lian, Batangas, Philippines, coll. Apr. 2010, 10 m.

**Museum repository:** Coral Museum, The Marine Science Institute, University of the Philippines

**Description.** The colonies may have a corallite formation that is flabello-meandroid (Fig. 5K) or phaceloid to phacelo-flabellate with short valleys that exhibit a meandering pattern. For example, specimen P1L02172 has a length, width, and height of approximately 90 mm, 67 mm, and 55 mm. Lengths of valleys in all specimens studied range from 18.95 mm to 47.95 mm and valley widths range from 4.70 mm to 18.95 mm. Septa occur in three to four orders in specimens with widths that range from 5.30 mm to 18.15 mm (Fig. 5L). The first order septa are typically exsert. Septa are granulated and shapes may be lobed, square, or angular, with smooth margins. Some specimens show a mix of the lobed and square-shaped septa, while other specimens exhibit an angular shape (Fig. 5M). The septa have relatively low angles that range from 12° to 55° from the horizontal plane of the valley (Fig. 5M). Columella are absent. The costae are lobed or blade-like protuberances that are mostly found at upper edge of the wall (Fig. 5N); otherwise the wall is smooth (Fig. 5O). A few of these protuberances are also found scattered in other parts of the wall away from the upper edge; otherwise, the wall is smooth.

Polyps are branched with knob-like tips that are fully extended during day time. They are typically translucent but may also be fleshy. Polyp color is commonly bright green or brown and the knob-like tips of the branches are usually white or bright green. The general appearance of the polyps is the basis for the common name "frogspawn". Colonies typically co-occur with *F. ancora* colonies along the reef slope. However, a majority of the *F. divisa* colonies are usually found in the deeper parts of the slope near the reef edge. They have been observed to thrive in silty and turbid environments. Those found in silty substrates are not usually attached or are weakly attached and are easy to collect.

**Comparisons with related taxa.** Polyps of *F. divisa* resemble *F. yaeyamaensis* in having branched polyps. Branches in the polyps of *F. divisa* are longer and widely-spaced in comparison with *F. yaeyamaensis,* which has branchlets that crowd along the main branch. Translucent polyps are a characteristic of *F. divisa* while, *F. yaeyamaensis* has fleshy, dark-colored polyps .The phacelo-flabellate coralla of *F. divisa* may be similar to the flabello-meandroid coralla of *F. ancora* and *F. yaeyamaensis.* However, their valleys are shorter compared with the valleys of *F. ancora* and *F. yaeyamaensis.* Septal angles of *F. divisa* are lower than *F. ancora. F. ancora* has the steepest angles amongst the three, followed by *F. divisa,* and then by *F. yaeyamaensis.*

**Occurrence:** Philippines (*Veron & Hodgson, 1989*; this publication), Malaysia (*Mazlan et al., 2005*; *Khodzori et al., 2015*), Australia (*Griffith, 2004*), Japan (*Veron, 1992*)

***Fimbriaphyllia yaeyamaensis*** (*Shirai, 1980*)
(Figs. 5P–5T)

*Botryphyllia yaeyamaensis Shirai, 1980*
*Euphyllia yaeyamaensis (Shirai, 1980)*

**Material studied:** (P1L02036), Clubhouse, Talim Bay, Lian, Batangas, Philippines, coll. Apr. 2008, <5 m (Figs. 5P, 5Q, 5S); (P1L02208), Clubhouse, Talim Bay, Lian, Batangas, Philippines, coll. May 2010, 11.4 m (Figs. 5R, 5T); (P1L02037, P1L02038), Clubhouse, Talim Bay, Lian, Batangas, Philippines, coll. Apr. 2008, <5 m; (P1L02186), Outer Talim, Lian, Batangas, Philippines, coll. Apr. 2010, 17 m.

**Museum repository:** Coral Museum, The Marine Science Institute, University of the Philippines

**Description.** Colonies may have a flabello-meandroid, phaceloid or phacelo-flabellate corallite formation. Colonies have short valleys with lengths that range from 24.85 mm to 87.55 mm with valley widths that range from 6.20 mm to 25.10 mm. Walls have been observed to form across valleys in specimens with longer valley lengths (Fig. 5Q). There are three orders of septa (Fig. 5Q) that primarily have square or angular shape with sharp or round corners (Fig. 5R). The lobe shape may still be observed in some specimens but a majority of the septa are angular in shape. Widths from the edge of the wall to the valley center range from 4.80 mm to 21.90 mm. The sides of the septa are granulated. The margins, generally, have a smooth appearance but may exhibit sharp corners down the slope. Septa may have relatively steep or low angles that depend on their position along the meandering valleys. Septa along the curved, round, or flaring sides of the valley have low angles compared with the septa that are directly on the opposite side and along straight valleys (Fig. 5R). Septal angles range from 8° to 55° (Fig. 5R). Columella are absent. The costae are in the form of long, prominent ridges that line the height of the wall (Fig. 5S). In some specimens, the costae are found only near the valleys or the upper part of the wall and the rest of the wall is smooth (Fig. 5T).

Live colonies have polyps with branchlets with knob-like tips. The branchlets are short and may be crowded along the main branch. Polyps are fleshy and are typically colored bright green or lime with white tips. Colonies are distributed along the shallow to the deep parts of the reef slope and co-occur with *F. ancora* and *F. divisa*.

**Comparison with related taxa.** Live colonies of *F. yaeyamaensis* resemble *F. divisa*. Polyps of *F. divisa* are translucent and the branches are usually long and widely-spaced, while colonies of *F. yaeyamaensis* have fleshy polyps with branchlets that crowd along the main branch. Flabello-meandroid colonies of *F. yaeyamaensis* have similarities with skeletons of *F. ancora* and flabello-meandroid skeletons of *F. divisa*. *F. yaeyamaensis* has lower septal angles compared with *F. ancora* and *F. divisa*. Walls forming across the valleys of *F. yaeyamaensis* are also unique to the species.

**Occurrence:** Philippines (*Veron & Hodgson, 1989*; this publication), Malaysia (*Mazlan et al., 2005*), Japan (*Veron, 1992*)

***Fimbriaphyllia paraancora*** (*Veron, 1990*)
(Figs. 5U–5Y)

*Euphyllia* sp.2 *Veron & Hodgson, 1989*
*Euphyllia paraancora Veron, 1990*; *Veron, 2000*

**Material studied:** (P1L02025), Malilnep, Bolinao, Pangasinan, Philippines, coll. May, 2008, <5.0 m (Figs. 5U, 5V, 5Y); (P1L02021), Malilnep, Bolinao, Pangasinan, Philippines, coll. May, 2008, <5.0 m (Figs. 5W, 5X).

**Museum repository:** Coral Museum, The Marine Science Institute, University of the Philippines

**Description.** Colonies exhibit a phaceloid corallite formation (Fig. 5U). The calices have diameters that range from 10.94 mm to 36.21 mm (Fig. 5V). Branch heights range from 68.29 mm to 90.62 mm from the main point of branching. Branches are wider near the calice and in parts along the branch where the costae are thicker. Calices are round to oblong (Fig. 5V). There are four orders of septa arranged with the second and third order in between the first order septa (Fig. 5V). The septa are lobed or are angular in shape and, generally, have smooth margins. Occasional wavy curves may be observed along the septal slope (Fig. 5V). The septa are not as exsert as in the septa of *F. ancora*. The septa curve down the center of the corallite creating a depression. Calices are deep and columella are absent (Fig. 5W). The costae are in the form of long, prominent ridges that line the length of the branch (Fig. 5X). Costae may be thickened near the calice and along the length of the branch, which manifests as a bulging ring around the branch (Figs. 5X–5Y). These thickenings may indicate prior positions of the calice along the branch during growth.

Live colonies have fleshy polyps that resemble the shapes of an anchor, kidney, or bean. Polyps are typically brown, pink, or green with the tips exhibiting the same color but may be shades lighter than the rest of the colony. Colonies inhabit the shallower parts of the reef slope but, although rare, may also be observed in the deeper parts of the reef slope.

**Comparison with related taxa.** Polyps of *F. paraancora* resemble the polyps of the flabello-meandroid colonies of *F. ancora*. Skeletons resemble the phaceloid colonies of *E. glabrescens* and *F. paradivisa*. Calices of *F. paraancora* have the largest diameter among the three phaceloid species. Calices of *F. paraancora* are also deeper compared with the calices of *E. glabrescens* and *F. paradivisa. E. glabrescens* generally exhibit costae in the form of fine striations and small lobes, while *F. paraancora* has long, prominent ridges as costae. *F. paradivisa* may have costae in the form of widely-spacsed ridges compared with the more crowded costae of *F. paraancora*.

**Occurrence:** Philippines (*Veron, 1990*; this publication), Taiwan (*Chen et al., 2005*);

***Fimbriaphyllia paradivisa*** (*Veron, 1990*)
(Figs. 5Z–5AD)

*Euphyllia* sp.1 *Veron & Hodgson, 1989*
*Euphyllia paradivisa Veron, 1990*; *Veron, 2000*

**Material studied:** (P1L02166), Reyna, Talim Bay, Lian, Batangas, Philippines, coll. Apr. 2010, 6 m (Figs. 5Z–5AC); (P1L02045), Clubhouse, Talim Bay, Lian, Batangas, coll. Apr.

2008, <5 m (Fig. 5AD); (P1L02182) Outer Talim, Lian, Batangas, Philippines, coll. Apr. 2010, 11m.

**Museum repository:** The Marine Science Institute, University of the Philippines

**Description.** Colonies exhibit a phaceloid corallite formation (Fig. 5Z). Calices may be round or oblate with diameters that range from 10.75 mm to 24. 35 mm (Fig. 5AA). Branch diameters range from 11. 85 mm to 20.10 mm. Branches have heights that range from 55.90 mm to 126. 80 mm from the main point of branching. There are three orders of septa (Fig. 5AA). The second and the third order septa are in between the first order septa. The first order septa are exsert and lobe-shaped with smooth margins. The septa are granulated on the sides. Septa are lobe-shaped to angular, with margins that generally have a smooth appearance (Fig. 5AB). Columella are absent. The costae occur as prominent ridges along the height of the branch. These are widely spaced from one another and sometimes have lobes (Figs. 5AC–5AD).

Live colonies have branched polyps with knob-like tips that are fully extended during day time. Polyps are usually green or brown with white tips. Colonies are distributed along the shallow and deep parts of the reef slope but they are most abundant in the shallow parts. The species has been reported to occur in abundance in mesophotic depths, i.e., beyond 30 m, in the Gulf Eilat, Red Sea, but are absent in the shallow areas (*Eyal et al., 2016*).

**Comparisons with related taxa.** Polyps of *F. paradivisa* resemble those of the phaceloid and flabello-meandroid colonies of *F. divisa*. *F. divisa* skeletons with the phaceloid corallite formation exhibit short valleys as opposed to round calices with small diameters, which are characteristic of *F. paradivisa*. Coralla of *F. paradivisa* resemble the phaceloid colonies of *F. paraancora* and *E. glabrescens*. *F. paraancora* has the largest calice diameter, while *F. paradivisa* has the smallest. However, in terms of branch height, *F. paradivisa* has the tallest colonies compared with *F. paraancora* and *E. glabrescens*. The shortest *F. paradivisa* among the samples is taller than the tallest *E. glabrescens* specimen in the collection (Fig. 6). The costal ridges of *F. paradivisa* are widely spaced compared with that of *F. paraancora*.

**Occurrence:** Philippines (*Veron, 1990*; this publication), Israel (*Eyal et al., 2016*)

## CONCLUSIONS AND RECOMMENDATIONS

The presence of two distinct and well-supported groups of *Euphyllia* in all the gene trees that retained the original members of the previous subgenera of *Euphyllia* and *Fimbriaphyllia* supports the recognition of *Fimbriaphyllia* as a genus. While the two subgenera previously represented a dichotomy that is based on colony structure, the dichotomy between the genera of *Euphyllia* and *Fimbriaphyllia* is now defined by polyp shapes, polyp length, sexuality, and mode of reproduction. The finer clustering of the 3′-end of the *cox1* gene exhibiting the distinct morphospecies of *Euphyllia* and *Fimbriaphyllia* shows that species are best identified through combining polyp morphology and colony structure. In the *Euphyllia* group, septal morphology appears to distinguish between species; thus, opening the possibility that there may be microstructures of the skeleton that may also be able to distinguish between species of *Fimbriaphyllia*.

The morphological characteristics and other biological attributes that were identified through the integrated approach on the, previously, largest genus in the Euphylliidae

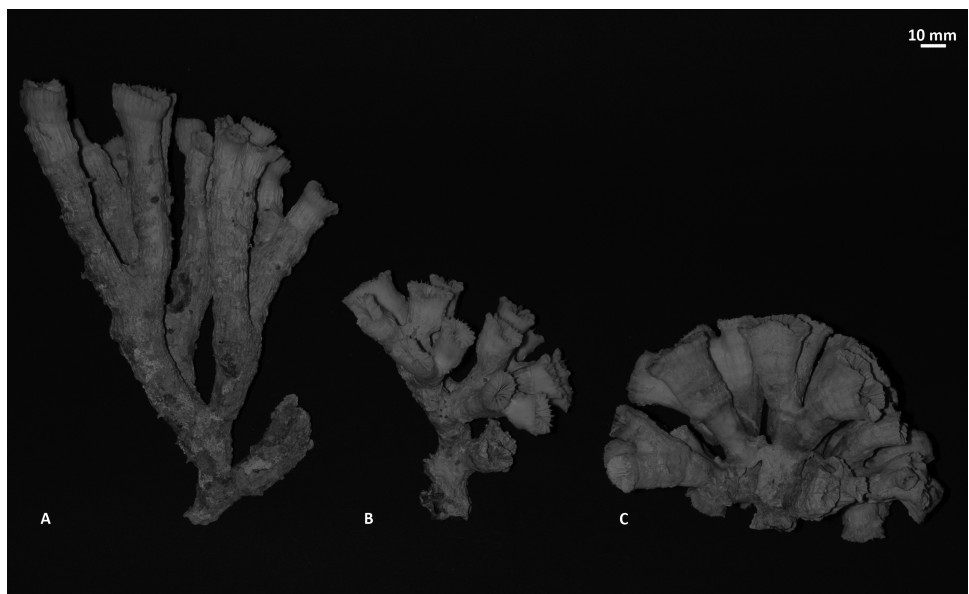

**Figure 6** **Height comparisons between *F. paradivisa* and *E. glabrescens*.** (A) Tallest and (B) shortest colonies of *F. paradivisa*. The shortest colony of *F. paradivisa*, among the specimen collection, is still taller than the tallest colony of (C) *E. glabrescens*.

(clade V) have shown patterns that are consistent with the other members of the clade as in *G. fascicularis* and *C. chagius* each diverged from the same ancestral node as *Fimbriaphyllia* and *Euphyllia* respectively. Apart from some similarities in skeletal traits, the polyp features that were identified for *Fimbriaphyllia* and *Euphyllia* were also found to be applicable to *G. fascicularis* and *C. chagius*. The pseudo-gynodioecious broadcast spawning trait of *G. fascicularis*, which was previously hypothesized as a transitional state between *Fimbriaphyllia* and *Euphyllia*, has been confirmed in this study. These findings show that the phylogenetic-based systematic scheme of *Euphyllia*, *Fimbriaphyllia*, *Galaxea*, and *Ctenella* are potentially operationally useful for the rest of clade V. Thus, laying the groundwork for fully resolving the phylogeny of the family Euphylliidae (clade V) as there are still species that still lack in molecular analyses. These species include two more species of *Euphyllia*, namely *E. paraglabrescens* and *E. baliensis*, which are of limited biogeographic range. *Gyrosmilia interrupta* and *Montigyra kenti*, species from the same family as *C. chagius* and also limited to the Indian Ocean, are also still unresolved. The inclusion of species of *Euphyllia*, *Fimbriaphyllia*, and *Galaxea* that are also found in the Indian Ocean and are within the geographic range of *C. chagius* is recommended for future studies.

## ACKNOWLEDGEMENTS

We are very grateful to Jean-Francois Flot and Danwei Huang for sharing their valuable knowledge and expertise on coral taxonomy and molecular biology and for their unwavering guidance all throughout the analyses. We would also like to thank them for lending us their time to provide initial comments and suggestions in the early stages of

the manuscript. We would like to thank Pei-Ciao Tang, Yaoyang Chuang, Sung-yin Yang, Meng I. Chen, Chai-Hsia Gan, Chieh Jhen Chen, Rose Lai, Hui-Wen Tung, Chao-Yang Kuo, Chia-Min Hsu, Jay Yang, and Shashank Keshavmurthy for providing invaluable laboratory support in Taiwan. We would also like to thank Dr. Porfirio Alino for his invaluable guidance and fruitful discussions in culling out the phylogeny of *Euphyllia.* We are, likewise, very grateful to Dr. Glenn G. Oyong and Dr. Esperanza C. Cabrera for sharing their valuable knowledge and expertise on laboratory techniques in Molecular Biology and Cellular Biology and for providing laboratory support in this entire endeavor. We would also like to thank Mr. Mark Windell Vergara, Mr. Ronald Violante, Mr. Benjamin Ermita, Ms. Maan Hontiveros, Mr. Alex Hontiveros, Mr. Ed Hontiveros, and Ms. Carina Escudero for the field support that they provided–for it is hard to collect corals without dive a buddy. We are also thankful to Dr. Irene Samonte-Padilla who also provided laboratory support and who helped us with the initial sample collection and tissue preservation on field. Special thanks to Dr. Robert Toonen, Dr. Zoe Richards, and two anonymous reviewers for their constructive comments that tremendously helped in improving the manuscript. Additional post-publication work would not have been possible without the guidance of Dr. Bert W. Hoeksema who graciously imparted his invaluable knowledge and insights on scleractinian taxonomy. We are, likewise, thankful to Dr. Esperanza Maribel Agoo for also sharing her knowledge and experience on systematics. We are also grateful to Ameurfina Koc and Jose Bernabe M. Magbanua for accommodating our use of their studio for the imaging of the coralla.

### Funding

This work was supported by the project of Wilfredo Roehl Y. Licuanan under the Coral Reef Targeted Research and Capacity Building for Management (CRTR) of the Global Environment Facility (GEF) funded by the World Bank, Academia Sinica Applications of the Thematic Research Program AS-99-TP-B22: Phylogenomic approaches to the evolutionary history of Family Acroporidae (Anthozoa; Scleractinia) of Chaolun Allen Chen, Ministry of Science and Technology Project of Chaolun Allen Chen NSC 96-2621-B-001-012-MY2 676116: Phylogenetic relationships of the Family Euphyllidae resolved through mitochondrial and nuclear genes, Kenting National Park Headquarters Project of Chaolun Allen Chen 673202-Long-Term Ecological Research (LTER) monitoring program, the NSF PIRE project OISE-0730256 of KE Carpenter, and the Monitoring and Impact Research on Resilience of Reefs (MIRROR) project of Dr. Wilfredo Roehl Y. Licuanan under the RESILIENT SEAS for Climate Change program funded by the Philippine Council for Agriculture, Aquatic and Natural Resources Research and Development (DOST-PCAARRD) formerly Philippine Council for Aquatic and Marine Research and Development (PCAMRD). Further work post-publication was also supported by the Synoptic Investigation of Human Impacts on Nearshore Environments (SHINE): Coral Reefs project of Dr. Wilfredo Licuanan under the National Assessment of Coral Reef

Environments (NACRE) program that was funded by the Department of Science and Technology- Philippine Council for Agriculture, Aquatic and Natural Resources Research and Development (DOST-PCAARRD). The funders had no role in study design, data collection and analysis, decision to publish, or preparation of the manuscript.

## Grant Disclosures

The following grant information was disclosed by the authors:
Coral Reef Targeted Research and Capacity Building for Management (CRTR) of the Global Environment Facility (GEF), World Bank.
Academia Sinica Applications of the Thematic Research Program: AS-99-TP-B22.
Ministry of Science and Technology Project: NSC 96-2621-B-001-012-MY2 676116.
Kenting National Park Headquarters Project 673202-Long-Term Ecological Research (LTER) monitoring program.
The NSF PIRE project: OISE-0730256.
Monitoring and Impact Research on Resilience of Reefs (MIRROR) project, RESILIENT SEAS for Climate Change program funded by the Philippine Council for Agriculture, Aquatic and Natural Resources Research and Development (DOST-PCAARRD) formerly Philippine Council for Aquatic and Marine Research and Development (PCAMRD).
Synoptic Investigation of Human Impacts on Nearshore Environments (SHINE): Coral Reefs project, National Assessment of Coral Reefs (NACRE) Program funded by the Department of Science and Technology- Philippine Council for Agriculture, Aquatic and Natural Resources Research and Development (DOST-PCAARRD).

## Competing Interests

The authors declare there are no competing interests.

## Author Contributions

- Katrina S. Luzon conceived and designed the experiments, performed the experiments, analyzed the data, wrote the paper, prepared figures and/or tables, reviewed drafts of the paper, collection of specimens.
- Mei-Fang Lin conceived and designed the experiments, performed the experiments, analyzed the data, contributed reagents/materials/analysis tools, wrote the paper, prepared figures and/or tables, reviewed drafts of the paper.
- Ma. Carmen A. Ablan Lagman conceived and designed the experiments, contributed reagents/materials/analysis tools, reviewed drafts of the paper.
- Wilfredo Roehl Y. Licuanan conceived and designed the experiments, contributed reagents/materials/analysis tools, reviewed drafts of the paper, collection of specimens.
- Chaolun Allen Chen conceived and designed the experiments, analyzed the data, contributed reagents/materials/analysis tools, wrote the paper, prepared figures and/or tables, reviewed drafts of the paper, collection of specimens.

## Field Study Permissions

The following information was supplied relating to field study approvals (i.e., approving body and any reference numbers):

A gratuitous permit to collect scleractinian corals was approved by the Bureau of Fisheries and Aquatic Resources (BFAR) of the Philippines (FBP-0021-08). The corals in Taiwan were collected through the permission granted by the Kenting National Park Authority as part of a long-term ecological monitoring program. As additional morphological work was performed post-publication, the MIRROR project helped us secure another gratuitous permit to collect more samples. The Bureau of Fisheries and Aquatic Resources (BFAR) gave us additional permission under the Gratuitous Permit No. FBP-0027-29.

### DNA Deposition

The following information was supplied regarding the deposition of DNA sequences:

All DNA sequences in this study are accessible via GenBank®. Please see the Supplemental File (Table S2) for the accession numbers.

### Data Availability

GenBank-Please see the supplemental file provided (Table S2).

### New Species Registration

The following information was supplied regarding the registration of a newly described species:

Publication: urn:lsid:zoobank.org:pub:EBF69BA0-897E-4AC8-ADF5-4A7115CA1353.

### Supplemental Information

Supplemental information for this article can be found online at http://dx.doi.org/10.7717/peerj.4074#supplemental-information.

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
