# Peer review of "Resurrecting a subgenus to genus: molecular phylogeny of Euphyllia and Fimbriaphyllia (order Scleractinia; family Euphyllidae; clade V)"

_PeerJ, doi:10.7717/peerj.4074_

## Round 0.1 · original submission · Major Revisions

· Academic Editor

Major Revisions

We now have feedback from 3 referees on your submission, and while supportive of the value of your study, two of them raise concerns about the taxon sampling in your study and the strength of the conclusions that can be drawn given the specific taxonomic choices included (or not) in your study. Both referees 1 & 3 comment on this issue being the justification for a recommendation of major revisions - either dialing back a little on the conclusions being drawn or at least specifically justifying the taxa that have been included in the study and including some caveats to how the results may change if taxon sampling was increased.

Beyond that one issue, which may not really be a major revision for you, I think the rest of the comments are relatively straightforward and should be simple to address. I look forward to seeing the revised manuscript.

·

Basic reporting

I applaud the authors for this well written and referenced manuscript. There are just a few places where the messages that were being conveyed were not clear. For example the last line of the conclusions in the abstract is difficult to understand - can you please revisit and reword that sentence.

Experimental design

The investigation has been performed to a high standard. Whilst additional within species replication would have been preferred and it would have been good to include other members of the Family Euphyllidae, nevertheless the results are meaningful. It would have been good to supplement the molecular work with some micromorphological analyses. Have the authors explored if there are morphological features beyond the colony shape that can help differentiate between species?

Furthermore, the data on soft tissues is pretty scant so I don't agree with pitching the paper this way.

Validity of the findings

The results that Euphyllia glabrescens is quite divergent to the other species is compelling. Also the finding that phaceloid colony structure is a convergent trait is also interesting.

I note the authors are choosing to discuss the clusters using subgenus names. However those subgenera have been synonomized, hence rather than continuing to use the subgeneric classification, I am wondering if the authors have considered whether E. glabrescens is the only true Euphyllia and whether the remaining species should be in the genus Fimbriaphyllia (aka elevate it from a subgenus). Considering the Euphyllidae clade is paraphyletic with Galaxea in the middle, it seems logical nevertheless I would appreciate if the authors can explain to me their rational for not doing this.

Can I also point out that in the Discussion paragraph of the abstract the first and last lines contradict themselves. Firstly you say the phylolgenetic trees show something different to the skeletal morphology, then at the end you say species within major groups can be distinguished by their colony structure - can you please revisit this.

Similarly in the Conclusions paragraph of the abstract it says in the first line that soft-tissue characteristics are needed to advance the goal of scleractinian classification but later on you say that Euphyllia is really the only genera that examining soft tissues (aka tentacles) is likely to work on.

Comments for the author

From my perspective, the value of the tentacles as a useful identification trait is well established, hence I do not think this study is adding anything new in that regard.... The authors emphasize the value of the tentacle length, however I question how the extent of tentacle inflation influences the tentacle length? If it does influence the length, then this would detract from its reliability as a character to delineate species. Hence while the title "when soft tissues talk" is quite catchy - and I agree with the premise that the best way to examine species boundaries is with an integrated approach - I question whether the soft tissues are providing the greatest insight into species level relationships in the Euphylila? The most robust data you present here is the molecular data - hence the title may be a bit of an over-stretch as it is currently written.

Please consider whether you would like to push the implications of this dataset further to classify species in Cluster A as belonging to the genus Fimbriaphyllia. If you reworked the paper this way, a change of title could be inevitable.

Reviewer 2 ·

Basic reporting

no comment

Experimental design

no comment

Validity of the findings

no comment

Comments for the author

Introduction, line 79: Recently, new two suborders for complex (including basal) and robust groups have been reported (Okubo 2016, Zoological Science 33:116-123). This may be referred.

Discussion, line 289-290: Veron (2000) does not use 'subgenus' in Euphyllia, and delete 'the subgenus' in this sentence.

Subgenera Fimbriaphyllia and Euphyllia have not been used in any other references.
And these subgenera were defined for only four species as below.
Euphyllia (Euphyllia) cristata Chevalier, 1971
Euphyllia (Euphyllia) glabrescens (Chamisso & Eysenhardt, 1821)
Euphyllia (Fimbriaphyllia) ancora Veron & Pichon, 1980
Euphyllia (Fimbriaphyllia) divisa Veron & Pichon, 1980
Therefore, if the authors resurrect these subgenera for other species, the authors should describe the definition for others newly.

Fig.2: one short branch of E. ancora P1L02014 is detached.

Fig. 3. It is hard to see the species relationship within clade A. The authors should make this difference clear in this tree.

Fig. 4: Veron (2000) does not use 'subgenus', so that left tree is not correct. In right tree, order of species should reflect the phylogeny of Fig. 3 (although it is not clear). For subgenus, please see above.

Table 1: in Group 2, colony form of E. yaeyamaensis should be phaceloid or fabellomeandroid. Corallites of E. yaeyamaensis and E. divisa should be rewritten (should focus on the corallite structure).

Reviewer 3 ·

Basic reporting

The basic reporting fulfills minimum standards.

Experimental design

The experimental design lacks sufficient phylogenetic and taxonomic sampling with regards to some of the claims; particularly polyphyly between genera.

Validity of the findings

Most of the findings are well supported however many of the conclusions and speculation need to be more carefully presented such that they are not overturned by additional work on this group.

Comments for the author

Review: When the soft tissues talk: molecular phylogeny of Euphyllia corals (Order Scleractinia; Family Euphyllidae; clade V)

General comments:
This study examines 3 molecular markers (cox1,cytb and B-tubulin) in Euphyllia species from the Philippines. The markers generally resolved two clear groups --although a portion of cox1 also appeared to be concordant with the nominal species as well. The two groups seemingly correspond to polyp morphology and to some degree life history traits, more than to the traditional morphological systematics which classifies the group based on the morphology of the coral colony (however there are only two main groups so I would caution against drawing too many conclusions). Another way of saying this is that colony morphology is more flexible than previously assumed in this group, which seems to be an emerging trend in molecular studies of Scleractinia. The study further claims that Euphyllia is paraphyletic with Galaxea, however; I think this claim needs more thought and evidence since the choice of outgroup could potentially influence this result. Pavona and Agaricia were chosen as outgroups and it is not made clear how distant these corals are, furthermore only the B-tubulin gene is used to compare these groups. If these corals are not closely related sister genera, then long branch attraction can cause improper rooting… if the trees are rooted with Galaxia for example then the group is no longer polyphyletic. I am not convinced that Pavona and Agaricia are the most appropriate outgroups and without additional taxonomic sampling it is hard to determine if they are the most appropriate or if there is polyphyly. Cox1 and cytB sequences should be much more prevalent in GenBank and if not then why was Galaxea not sequenced with these markers? Why not examine with many possible outgroups to determine the most appropriate rooting? The figures should be labeled as A and B, etc. for multipannel figures and referred to in the legend (and all fonts should be large enough to be clearly readable). The figures could also be improved by adding some of the graphics from figure 4 next to the trees, to illustrate the range of colony and polyp morphology in each group for example, without the need to skip back and forth between figures. Overall the study represents a first pass and very preliminary look at this group, which is important; however the paper needs to be rewritten to present itself as such a preliminary look so as to lay a foundation for future work. Given the limited phylogenetic and taxonomic sampling of this study I would warn against making too many claims that maybe contradicted later. I recommend a major revision or resubmission.

Specific comments:

Line 37; change nominated to nominal
Line 39; I would say that several markers were examined but a multilocus approach might be stretching it a bit.
Line 50: change taxonomic to phylogenetic or systematic (taxonomy only refers to names and not relationships.
Line 57; Species can then be distinguished by their colony structure? After what? Please clarify.

Methodology
Line 138: delete gratuitous
Line 192: by slicing, do you mean gel excision?
Line 217: it would be good to mention Table S2 for the first time at the end of this sentence
Line 229; it would be good to mention the rationale behind choosing Pavona and Agaricia as outgroups… where there no closer sequences available for the chosen genes? If the outgroups are too far, then long branch attraction may influence the order and branching patterns within the clades…
Line 240; what about other possibilities like introgression?

Line 258; replace although with however.
Line 332 and 335; replace taxonomic with systematic

Figure 1
Where is clade B in the combined figure? Could the sequences from genbank not be concatenated as well? I suggest explaining in the legend. The type in the insets is too small, very hard to read co1 and ctyB. also I suggest adding labels to each inset (eg. A, B,C ) and explain in the legend, eg. A combined tree, B) cox1 alone, C) cytb alone. The type in the scale bar is too small to read (and the line a bit too thin)

---

## Round 0.2 · Minor Revisions

· Academic Editor

Minor Revisions

I am sorry for the delay in getting this back to you, but we have been unable to get feedback from the first referee on the submission, so we are going to move forward with the comments from referees 2 & 3 alone. Both appreciated the detailed revisions and are supportive of the paper moving forward into publication. I am returning your manuscript for (very) minor revisions so that you are able to make a simple correction and a decision on whether or not you wish to follow the suggestion of the third referee.

The second referee asks you to revise your reference to Lin et al. (2011) for accuracy, which seems a reasonable request. The third referee sees no major issues with moving the paper forward but suggests that a redrawn tree (or even cartoon of the major result) from Fukami et al. (2008) would be helpful to put this work into context. I find myself agreeing with them that it would help me to better follow the text to have that figure available here rather than pulling up the original paper. However, I also find myself in agreement with the referee that this suggestion is for convenience of the reader, and should not be a condition for acceptance. Further, we certainly do not want the entire multi-panel tree from that paper reproduced here, so I leave it to the authors as to whether or not they feel that inclusion of a simple figure highlighting the most important details of that tree (or even just clades IV & V) for this analysis as an additional figure.

Reviewer 2 ·

Basic reporting

no comment

Experimental design

no comment

Validity of the findings

no comment

Comments for the author

Totally this ms was revised properly following the review comments.
I have only one thing to be revised.
Lin et al. (2011), which the authors also referred in the text, represented a phylogenetic relationship of Euphyllia spp. in Fig. 6. Actually, this tree in Lin et al. (2011) also showed two groupings of Euphyllia spp. and one of two clades includes E. glabrescens and E. paraglabrescens. The authors should refer this data in the text (discussion part, especially).

Reviewer 3 ·

Basic reporting

No comment

Experimental design

No Comment

Validity of the findings

No Comment

Comments for the author

The authors have gone to great length to meet all of concerns of the reviewers and the revised manuscript is now acceptable for publication in PeerJ. The only non-compulsory suggestion that I have for improvement is to include a redrawn figure or diagram placing this study in more broad phylogenetic context (e.g. a cartoon diagram of a portion of Fukami's 2008 tree showing the major clades and approximate placement of Euphyllia)... even as a supplementary figure, this might help refer the unfamiliar reader to some of the issues mentioned in the introduction and discussion, and may simplify some of the contextual explanations. This is the only suggestion that I have and even without this addition, the manuscript is suitable for publication.

---

## Round 0.3 · accepted · Accept

· Academic Editor

Accept

I have read your response, and believe that the referee concerns have all been addressed. Given that the feedback on the last version was relatively minor, I see no reason to delay this any longer, and am happy to move this forward into production.